# Learnability and Privacy Vulnerability are Entangled in a Few Critical Weights

**Xingli Fang & Jung-Eun Kim**[*]
Department of Computer Science
North Carolina State University
Raleigh, NC 27695, USA
{xfang23, jung-eun.kim}@ncsu.edu

## Abstract

Prior approaches for membership privacy preservation usually update or retrain all weights in neural networks, which is costly and can lead to unnecessary utility loss or even more serious misalignment in predictions between training data and non-training data. In this work, we observed three insights: i) privacy vulnerability exists in a very small fraction of weights; ii) however, most of those weights also critically impact utility performance; iii) the importance of weights stems from their locations rather than their values. According to these insights, to preserve privacy, we score critical weights, and instead of discarding those neurons, we rewind only the weights for fine-tuning. We show that, through extensive experiments, this mechanism exhibits outperforming resilience in most cases against Membership Inference Attacks while maintaining utility.

## 1 Introduction

Membership privacy risks of machine learning models arise from models' behavioral discrepancy between training and non-training data points. Leveraging such a discrepancy, an attacker can discriminate membership information whether a data point was used for training the victim model Shokri et al. (2017). This attack model is called membership inference attacks (MIAs). Existing studies Carlini et al. (2022b); Ye et al. (2024) pointed out that some data points are more privacy-vulnerable than others. Li et al. (2024) suggested that better privacy-utility can be achieved by focusing on these data points. However, privacy-preserving training on the model-end is still in a black-box stage. On the other stream of work, early studies Frankle & Carbin (2019); Molchanov et al. (2019); Lee et al. (2019) have shown that a subnetwork existing in a neural network can achieve competitive performance, identifying that only a lesser fraction of weights contributes to the model's utility. These prior studies collectively motivate us to raise a reflective question: *Do there exist only some weights whose updates lead to privacy leakage of learning models?*

To locate them, we first propose a weight-level importance estimation based on Machine Unlearning (MU) to measure fine-grained privacy vulnerability existing in neural networks. With our approach, we find that weights that cause the model to be privacy-vulnerable are only present in a small fraction of the weights. Moreover, we observe that a large portion of these weights overlaps with the learnability-critical weights. It explains why Yuan & Zhang (2022) fails to mitigate privacy risks using general pruning techniques.

One of our very important observations is that the importance of weights—in terms of accuracy—stems from their locations rather than their values. As long as the most critical weights (the proportion can be even down to $0.1\%$) remain in the model—i.e., are not pruned or removed—and rewind them in their initial values, the model can recover its accuracy even when these weights are left unupdated after retraining or fine-tuning. Building on top of these insights, we design a fine-tuning strategy that curates only privacy-vulnerable weights. To the best of our knowledge, our approach is the first to perform membership-privacy-oriented fine-tuning at a weight-level granularity. Through comprehensive experiments against modern membership inference attacks, LiRA and

---

[*]Corresponding author

RMIA Carlini et al. (2022a); Zarifzadeh et al. (2024), we demonstrate that, in terms of privacy-utility tradeoffs, our strategy outperforms existing privacy-defending methods that train machine learning models even from scratch. We emphasize the following core insights that we identified through this paper: (*i*) Privacy vulnerability exists in a **very small** fraction of weights. (*ii*) However, most of those weights **also** critically impact utility performance. (*iii*) The importance of weights stems from their **locations** rather than their values.

## 2   PRELIMINARIES AND RELATED WORK (MORE CONTINUED IN APPENDIX)

In this section, we introduce fundamental background knowledge regarding Membership Inference Attack, and prior studies regarding Importance estimation of components in neural networks. Due to page limitations, further related work concerning Membership privacy preservation methods and machine unlearning is presented in Appendix A.

### 2.1   INTRODUCTION TO MEMBERSHIP INFERENCE ATTACKS

In our study, we focus on membership privacy on classification tasks. In Membership Inference Attacks (MIAs), the attacker's goal is to determine whether a given sample was part of the training dataset of a target (or victim) model. Formally, consider a target model, $f(\cdot; \theta) : \mathbb{R}^{C_{in}} \to \mathbb{R}^{C_{out}}$, where $C_{in}$ is the input dimensionality and $C_{out}$ is the class count of the task. A membership inference attack can be formulated as

$$\mathcal{A} : f(\boldsymbol{x}; \theta) \to \{0, 1\}, \tag{1}$$

where $\mathcal{A}$ is a binary classifier that outputs 1 if the sample $\boldsymbol{x}$ is inferred to be a member of the training set of $f(\cdot; \theta)$, and 0 otherwise. The design of the attack function $\mathcal{A}$ depends heavily on the attack strategy. In neural network (NN)-based MIAs Shokri et al. (2017); Salem et al. (2019), $\mathcal{A}$ itself is a machine learning model trained on the predictions of the target model. In contrast, in metric-based approaches (e.g., threshold-based MIAs) Song & Mittal (2021); Del Grosso et al. (2022); Carlini et al. (2022a); Leemann et al. (2023); Zarifzadeh et al. (2024), $\mathcal{A}$ is defined by a manually specified function that computes certain statistics (such as confidence scores or loss values) and compares them against a threshold, typically chosen using auxiliary techniques such as shadow models Shokri et al. (2017); Carlini et al. (2022a).

### 2.2   IMPORTANCE ESTIMATION OF COMPONENTS IN NEURAL NETWORKS

The importance estimation of components in neural networks has mainly been studied in the context of model pruning. Frankle & Carbin (2019) observed that the potential of weights can be determined, in terms of generalizability, once the model is initialized. Lee et al. (2019); Molchanov et al. (2019) made use of weight gradients in searching for subnetworks with comparable generalizability to the original model. Liebenwein et al. (2021) explored possible loss beyond generalizability in pruning. Ye et al. (2019); Sehwag et al. (2020) explored how to prune neural networks in the adversarial environment. Tang et al. (2020) assessed the reliability importance of neurons by aligning spurious and clean samples through learnable masks. Frankle et al. (2020) observed that weight rewinding helps fine-tuning of extremely sparse models. Renda et al. (2020) found fine-tuning with rewound weights usually outperforms direct (*a.k.a.*, in-place) fine-tuning. Gadhikar & Burkholz (2024) analyzed the factors why learning rate rewinding, along with weight rewinding, recovers utility better. Tran et al. (2022) found that models suffer from fairness deterioration after pruning. Wang et al. (2023) computed connectivity importance via the influence on the spectrum of the neural tangent kernel (NTK) Jacot et al. (2018). Jia et al. (2023) found machine unlearning can benefit from magnitude pruning. Sun et al. (2024) applied activation into importance estimation based on the characteristics of large language model. Ye et al. (2025) proposed a training-free importance estimation and pruning on foundation models. Our work is distinct in that we identify privacy-vulnerability of weights.

## 3   MOTIVATION: REMOVING UNIMPORTANT WEIGHTS IS INEFFECTIVE FOR PRIVACY

One of the fundamental weight/neuron importance estimation methods is Taylor First Order (TFO) Molchanov et al. (2019). The method estimates the global weight importance via magnitudes of

gradients and weights, which is formulated as follows:

$$S = \{s_i\}_{i=1}^m = \{ \sum_{d \in D_{str}} |g_{i,d} w_{i,d}| \}_{i=1}^m \tag{2}$$

where $S$ denotes the set of importance scores of weights in the evaluated model, $s_i$ denotes the importance score of the weight, $w_i$, $w_{i,d}$ denotes the value of the $i$-th weight of the model before updating with the data point $d$, $g_{i,d}$ denotes the $i$-th weight's gradient computed under data point $d$, $D_{str}$ denotes the randomly selected subset of training data $D_{tr}$ (i.e., $D_{str} \subseteq D_{tr}$), and $m$ denotes the number of weights the model contains. In TFO, the approach usually accumulates the scores in tens of iterations along with the model update in each turn of filter removals of the model. Although the TFO groups weight scores into their belonging filters/neurons ultimately for filter/neuron pruning, we use the primitive weight scores for one-shot weight-level pruning.

In detail, to identify the most critical weights, according to the importance estimation method, we prune out the least important weights in one shot instead of iterative and gradual removal as in the original TFO. Figs. 1a and 1b exhibit that, even in the very high sparsities, accuracy is maintained, but privacy vulnerability does not improve. Also, at times, the model becomes even more vulnerable after pruning, evidenced by the increase of the testing loss of 90% sparsity from 0% one (non-pruned) as shown in Fig. 1b, and also the observation by Yuan & Zhang (2022) that MIAs on some pruned models become more successful. Overall, these observations lead us to conjecture that,

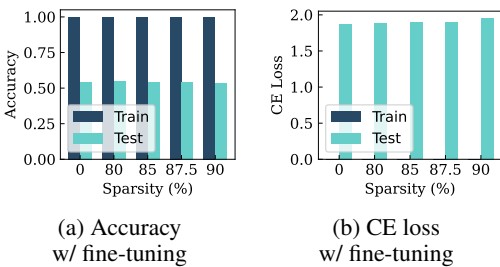

(a) Accuracy
w/ fine-tuning

(b) CE loss
w/ fine-tuning

Figure 1: According to TFO, important weights are pruned over different sparsities. The results are shown on ResNet18 and CIFAR-100

> Conjecture: *The performance impact and privacy vulnerability are entangled and exist in a very small number of weights.*

An intuitive way for verifying this conjecture is to show a correlation between privacy vulnerability and performance impact. For the goal, we distinguish the traditional estimation of how to maintain utility performance from the estimation of privacy vulnerability. We here refer to the importance estimation for utility performance (i.e., accuracy) in the common pruning techniques as *learnability* while we refer to how privacy-vulnerable a weight can become as *privacy vulnerability*. In the next section, we first propose our approach to estimate privacy vulnerability. Then, the entanglement issue of learnability and privacy vulnerability is empirically shown, and we discuss how to solve it.

## 4 PROBLEM SETUP AND METHODOLOGY

### 4.1 PRIVACY VULNERABILITY ESTIMATION

Membership privacy vulnerability is mainly due to the behavioral disparity between member and non-member data. Hence, the intuition of our approach is to determine critical weights of the model that exacerbate the discrepancy between the two prediction distributions to preserve privacy. To achieve this goal, we make use of the concept of machine unlearning Bourtoule et al. (2021) to design a mechanism to let the model **learn member data** while **unlearning non-member data**, respectively.

Our privacy vulnerability estimation approach (Fig. 2b) consists of a unprotected model, $M_{up}$; a vanilla model, $M_{vn}$; member set, $D_{tr}$; and non-member set, $D_{re}$. The $D_{tr}$ is the set on which the $M_{up}$ is trained. The non-member set, $D_{re}$, is a held-out set of data points that the $M_{up}$ has never seen during training, and it is also disjoined from the testing data in the evaluation phase. The two models, $M_{up}$ and $M_{vn}$, are in the same structure, $f(\cdot; \theta)$, but with different parameters, $\theta_{up}$ and $\theta_{vn}$, respectively. $\theta_{up}$ are pretrained on training data $D_{tr}$ while $\theta_{vn}$ are the values at initialization before being trained on $D_{tr}$.

For member data, $D_{str}$, we force the model to minimize the loss as much as possible. In contrast, for non-members, $D_{sre}$, we encourage the predictions close to the vanilla model, $M_{vn}$, rather than ground truths. This process can be formulated as follows:

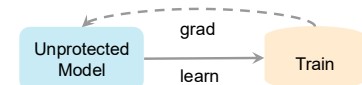

(a) Existing learnability estimation

$$\arg\min_{\theta_{up}}\{\mathbb{E}_{(x,y)\sim D_{tr}}\left[\mathcal{L}_{\text{ce}}(x,y;M_{up})\right],$$
$$\mathbb{E}_{(x,y)\sim D_{re}}\left[\mathcal{L}_{\text{kl}}(x;M_{up},M_{vn})\right]\} \quad (3)$$

where $\mathcal{L}_{ce}$ denotes the cross-entropy loss function, and $\mathcal{L}_{kl}$ denotes Kullback-Leibler (KL) divergence Csiszár (1975); Hinton et al. (2015). Through this process (Eq. 3), the model tries to learn information that is only effective for recognizing member data points so that it can maintain low loss on the train set when unlearning the non-member set, which does not contribute to the privacy vulnerability of the model since the data points are all non-member. In details, we fine-tune the unprotected model, $M_{up}$, using the following objective function:

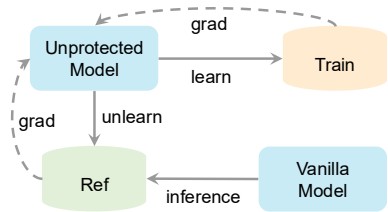

(b) Our proposed privacy vulnerability estimation

Figure 2: Our approach takes into account privacy vulnerability for importance estimation, while TFO only measures learnability for accuracy.

$$\mathcal{L}_{\text{pve}} = (1-\lambda)\mathcal{L}_{\text{ce}}(f(x_{tr};\theta_{up}),y_{tr}) + \lambda\mathcal{L}_{\text{kl}}(f(x_{re};\theta_{up}),f(x_{re};\theta_{vn})) \quad (4)$$

where $(x_{tr},y_{tr})$ and $x_{re}$ are mini-batch samples randomly sampled from $D_{tr}$ and $D_{re}$, respectively; $\lambda$ is hyper-parameter to balance the learning and unlearning losses so that the fine-tuned model can maintain accuracy on $D_{tr}$ while losing accuracy on $D_{re}$ as much as possible. The final privacy vulnerability estimation function is the same as Eq. 2 but with these aforementioned processes and constraints. It accumulates the weight-level importance with respect to privacy vulnerability, via gradients and magnitudes at each step, along with the update of $\theta_{up}$.

## 4.2 LEARNABILITY AND PRIVACY VULNERABILITY ARE ENTANGLED

To verify our conjecture in Sec. 3, we visualize the weight-level privacy vulnerability scores and learnability scores in Fig. 3 and quantify their correlations in Tab. 1 with two architectures: ResNet18 He et al. (2016) and ViT Dosovitskiy et al. (2021). Shown by the charts for all trainable weights (the leftmost column) in Fig. 3, most of the weights are neither privacy-vulnerable nor learnability-critical, which aligns with the experimental results in Fig. 1. It tells again that pruning

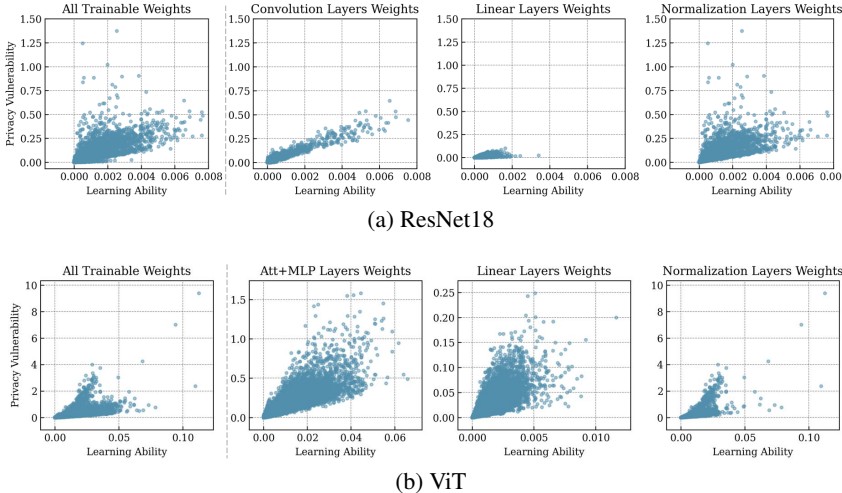

(a) ResNet18

(b) ViT

Figure 3: The visualization of weight-level learnability scores and privacy vulnerability scores. Privacy vulnerability and accuracy are significantly correlated and this correlation varies in different components. Due to the significant scale discrepancy, the ranges of axes of the four charts in ViT are not consistent. (The same data points as Tab.1.)

learnability-noncritical (not critical for accuracy) weights does not remove the privacy risks (prediction discrepancy).

The other weights, much fewer than these non-critical weights, can be categorized into three types: privacy-vulnerable, learnability-critical, and both. Tab. 1 shows the Pearson correlation coefficient between privacy vulnerability and learning ability. We find that the results of the two architectures are consistent that the correlation in normalization layers (batch normalization Ioffe & Szegedy (2015) in ResNet18 and layer normalization Ba et al. (2016) in ViT) are the lowest while the correlation in main components of the models (convolution layers in ResNet18 and Attention & MLP layers in ViT) are the highest. Weights belonging to normalization layers occupy only a tiny proportion of weights–less than $1\%$. However, some of them are the highly privacy-vulnerable weights of the models as shown in the charts of normalization layers weights (the 3rd column) in Fig. 3. Since these weights are also critical for learnability (many weights in normalization layers exhibit high learnability scores), pruning them by common pruning techniques will impair the performance.

Moreover, the majority of the weights belong to convolution/attention/MLP layers, and they show strong correlations—over $0.9$ in Pearson correlation coefficient—between privacy-vulnerability and learnability (see Tab. 1). The correlations are significantly higher than normalization layers. This result indicates that many privacy-vulnerable weights are also crucial for learnability. In addition, compared to CNNs, transformers exhibit higher privacy vulnerability (see charts of convolution layers weights and Att+MLP layers weights (2nd column in Fig. 3)), which is also supported in part by the observation of Zhang et al. (2024) that attention layers lead to worse privacy risks.

Table 1: The correlation between privacy vulnerability and learnability in two architectures. PCC denotes Pearson Correlation Coefficient. `Att+MLP` denotes the weights of the attention layers and MLP layers in transformer blocks. (The same data points as in Fig. 3b)

| Model | Weight Type | PCC | Proportion |
|---|---|---|---|
| ResNet18 | All | 0.8329 | 100.00% |
| | Conv | 0.9410 | 99.50% |
| | Linear | 0.8096 | 0.45% |
| | Norm | 0.6776 | 0.05% |
| ViT | All | 0.7667 | 100.00% |
| | Att+MLP | 0.9068 | 99.39% |
| | Linear | 0.8642 | 0.54% |
| | Norm | 0.7336 | 0.07% |

Finally, the linear layers in Tab. 1 denote the last few linear layers. We find that most weights in them are not privacy-vulnerable, while some of them could be learnability-critical.

In summary, **most privacy-vulnerable weights impact learnability** (utility performance). This is the fundamental reason why the existing standard pruning techniques fail to effectively reduce privacy risks. To address this issue, we propose **C**ritical **W**eights **R**ewinding and **F**inetuning (**CWRF**) in the next section to promote the model to achieve better privacy-accuracy trade-offs.

## 4.3 CRITICAL WEIGHTS REWINDING AND FINETUNING (CWRF)

Our approach (CWRF) consists of three steps: (*i*) estimating privacy vulnerability, (*ii*) rewinding & freezing privacy-vulnerable weights, and (*iii*) fine-tuning the other weights with privacy-preservation training approaches. Since privacy vulnerability estimation has been elaborated in Sec.4.1, we start our discussion from the second step.

**Weights Rewinding.** Weights rewinding Renda et al. (2020); Frankle et al. (2020) is a strategy that rolls back weights to earlier values in training. In our approach, the weights are rewound to the initial status, at which point the weights are privacy-safe because no data has been exposed to the model. Once calculating the privacy vulnerability estimation scores $\mathcal{S}_{pve}$ in the way described in Sec.4.1, two masks for weights rewinding and fine-tuning can be produced as follows:

$$\mathcal{B}_r = \{\mathbb{I}[s_i \geq Q(S_{pve}, r)]\}_{s_i \in S_{pve}}, \quad \mathcal{B}_f = 1 - \mathcal{B}_r \tag{5}$$

where $\mathcal{B}_r$ denotes weight rewinding mask, $\mathcal{B}_f$ denotes weight freezing mask, $\mathbb{I}(\cdot)$ denotes indicator function, $Q(\cdot, \cdot)$ denotes the combination of sort function in descending order and quantile function, and $r$ denotes the predefined rewinding rate we opt to. After producing the masks, a portion of the weights of the trained model is rewound from $\theta_{up}$ to $\theta_{vn}$ (defined in Sec.4.1) as follows:

$$\theta_{rw} = \mathcal{B}_f \odot \theta_{up} + \mathcal{B}_r \odot \theta_{vn} \tag{6}$$

---

**Algorithm 1:** Pseudocode of CWRF

---

**Input:** Unprotected model $M_{up}$ with parameters $\theta_{up}$, vanilla model $M_{vn}$ with parameters $\theta_{vn}$, member (train) set $D_{tr}$, and non-member (reference) set $D_{re}$, batch size $B$, privacy-preserving training approach $\mathcal{P}$, the number of iterations for score estimation $T$, the number of fine-tuning epoches $E$, the learning rate for estimation $\eta_e$, the learning rate for fine-tuning $\eta_t$.

**Result:** Privacy-fine-tuned $M_{up}$ with parameters $\theta_{up}$

1   Initialize $\{\phi_j = 0\}_{j=1}^{N}$ which are corresponded to weights of $\theta_{up}$

2   Copy unprotected model, denoted as $M_{up}'$ with parameters $\theta_{up}'$

3   **for** $i = 1 \ldots T$ **do**

4      Get sample batches $\{(x_i^{tr}, y_i^{tr})\}_{i=1}^{B} \subset D_{tr}$ and $\{(x_i^{re}, y_i^{re})\}_{i=1}^{B} \subset D_{re}$

5      Forward and compute loss $\mathcal{L}_{pve}(M_{up}'(x_i^{tr}), y_i^{tr}, M_{up}'(x_i^{re}), M_{vn}(x_i^{re}))$

6      (For $\mathcal{L}_{pve}$, refer to Eq. 4)

7      Approximate gradient $\mathcal{I} \leftarrow \nabla_{\theta_{up}'} \mathcal{L}_{pve}$

8      Compute scores $\phi \leftarrow \phi + |\mathcal{I}\theta_{up}'|$     (refer to Eq. 2)

9      Update unprotected model $\theta_{up}' \leftarrow \theta_{up}' - \eta_e \mathcal{I}$

10   **end**

11   Get the two masks $\mathcal{B}_r = \{\mathbb{I}[s_i \geq Q(S_{pve}, r)]\}_{s_i \in S_{pve}}, \mathcal{B}_f = 1 - \mathcal{B}_r$    (refer to Eq. 5)

12   Rewind the unprotected model $\theta_{up} \leftarrow \mathcal{B}_f \odot \theta_{up} + \mathcal{B}_r \odot \theta_{vn}$    (refer to Eq. 6)

13   **for** $epoch = 1 \ldots E$ **do**

14      **for** $i = 1 \ldots K$ **do**

15          (K denotes the number of mini-batches)

16          Get sample batches $\{d_i^{tr} = (x_i^{tr}, y_i^{tr})\}_{i=1}^{B} \subset D_{tr}$

17          (Some preserving approaches may additionally require reference data)

18          Train the unprotected model with privacy approach $\mathcal{P}(M_{up}, d_i^{tr})$

19          Approximate gradient $\mathcal{I} \leftarrow \nabla_{\theta_{up}} \mathcal{P}$

20          Update the model $M_{up}$ with masks $\theta_{up} \leftarrow \theta_{up} - \eta_t \mathcal{I} \mathcal{B}_f$    (refer to Eq. 7)

21      **end**

22   **end**

---

where $\odot$ denotes Hadamard product and $\theta_{rw}$ is the updated weights with partially rewound weights after the two masks are overlaid. After rewinding, the most privacy-risky weights can return to being privacy-safe. However, due to entanglement between privacy-vulnerability and learnability, the rewinding also leads to the utility deterioration of the model. More precisely, it usually leads to random-guess-level utility. Hence, the model needs to be fine-tuned to recover its utility.

**Weights Freezing & Privacy Fine-Tuning.** The final step is fine-tuning the model to achieve better privacy-utility trade-offs. It consists of two parts: Weights freezing & privacy fine-tuning. For training $\theta_{rw}$ to preserve privacy, we can plug in any privacy-preserving approaches and train the model. Note that the approaches need to train the model from scratch, but by being plugged into our method, they only require partial weights to be rewound and frozen, and then the rest of the weights are fine-tuned. From the perspective of implementing weight freezing, masking the gradients is a sensible option to stop the update of the non-rewound weights. Given the gradients, $\mathcal{G}_p$, obtained by the privacy-preserving training approach with the rewound weights, $\theta_{rw}$, at each fine-tuning iteration, we can filter out the gradients of the frozen weights so that only the rewound weights can be updated:

$$\mathcal{G}_p \leftarrow \mathcal{B}_f \odot \mathcal{G}_p \tag{7}$$

During the fine-tuning process, we do not train a model at a fixed learning rate because neither a too small or too large fixed learning rate is good at recovering the model from random guess status. Instead, the learning rate is also rewound to the earliest learning rate at which the model started. The way is similar to learning rate rewinding (LRR) Frankle et al. (2020); Gadhikar & Burkholz (2024), although we rewind the learning rate to the very initial one. The self-contained procedure of CWRF is described in Alg. 1. The CWRF contains three stages: (*i*) scoring privacy vulnerability, (*ii*) rewinding and freezing privacy-vulnerable weights according to scores, and (*iii*) fine-tuning the

rest of the trainable weights with a privacy-preserving approach. CWRF can adapt arbitrary privacy training approaches by plugging them into the third stage of CWRF for privacy-post-training. We note that it might be somewhat counterintuitive to fine-tune the privacy-invulnerable weights rather than the privacy-vulnerable. There are two reasons why the model is fine-tuned that way: (*i*) the privacy risks of the privacy-vulnerable weights have been fully removed thanks to rewinding. Fine-tuning the rest of less- or in-vulnerable weights help the model with further mitigation of privacy risks. (*ii*) based on our hypothesis and empirical investigation elaborated and explained in Sec. 4.4, fine-tuning privacy-invulnerable weights help the model recover its utility better than doing that on privacy-vulnerable weights. We explain this in detail in the next section.

## 4.4 THE PRIVACY-VULNERABLE WEIGHTS ARE UNNECESSARY TO BE TRAINED

Finally, we explain why we fine-tune the privacy-invulnerable weights rather than the vulnerable. The lottery hypothesis Frankle & Carbin (2019) proposed and validated that the learnability of weights in a neural network is determined at the initialization phase. Motivated by the insight, we propose and validate a hypothesis in this section:

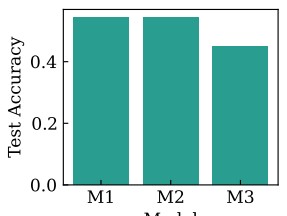

> Hypothesis: *The learnability of a weight in a neural network is determined by its position rather than its value (magnitude & sign.)*

This can be observed and understood through model pruning. For the verification, we devised three models:

Figure 4: The performance of M1, M2, & M3 on ResNet18 & CIFAR-100.

- M1: unpruned model trained from scratch.
- M2: 85% pruned model from M1 and then rewound to the initial values and retrained.
- M3: 85% pruned model from M1 with no fine-tuning/retraining

M2 and M3 are pruned with the same masks based on M1. Their comparisons are shown in Fig. 4. Let us focus on the learnability-unimportant weights that are present in M1 (which are pruned away in M2 and M3.) By looking at the almost same final accuracy of M1 and M2, we can infer that in M1 the learnability-unimportant weights shared knowledge and role with the learnability-important weights. This is also cross-checked by the accuracy drop of M3 (from M1) where the learnability-unimportant are discarded. It hints at the potential of the pruned weights (which were regarded as not important for learnability, though) toward learnability to some extent. Overall, it is encouraged not to update learnability-important weights by the Hypothesis, but to finetune learnability-unimportant weights by Fig. 4. On top of that, by considering that privacy-vulnerable weights are entangled with learnability-critical weights, we only rewind the privacy-vulnerable weights so as not to hurt the accuracy, but fine-tune only privacy-invulnerable weights - not to expose the privacy-vulnerable weights to the data again to reduce privacy risk.

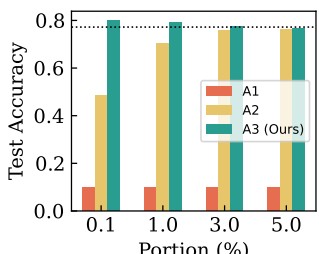

Figure 5: The performance of A1, A2, & A3 along with removing/rewinding ratios. The dotted line represents a baseline performance of a model trained from scratch with the same privacy-preserving approach

Based on the insights, to verify the hypothesis and validate our approach, CWRF, we compare the following three approaches:

- A1: Remove privacy-vulnerable weights & fine-tune privacy-invulnerable weights;
- A2: Rewind privacy-vulnerable weights & fine-tune privacy-vulnerable weights;
- A3 (CWRF): Rewind privacy-vulnerable weights & fine-tune privacy-invulnerable weights.

As for privacy-preserving training, here we apply RelaxLoss Chen et al. (2022) to fine-tune the three approaches. Shown in Fig. 5, it is very clear that discarding privacy-vulnerable weights (A1) leads to unrecoverable accuracy crash for the model, unlike the cases of A2 & A3. The performance discrepancy stems from "removing" (A1) vs. "rewinding" weights (A2 & A3). That is because

removing alters the locations of the weights, but rewinding does not. This comparison successfully validates our hypothesis that the locations of weights are of paramount importance for learnability.

As long as the crucial locations in the model are retained, the model preserves the capability to recover its accuracy. Another point to pay attention to is the performance gap between A2 and A3. By retaining the locations of privacy-vulnerable weights (A3), the model can recover its accuracy when a very small portion of privacy-vulnerable weights are rewound, and it even outperforms the baseline model that is trained from scratch using RelaxLoss with the same training configurations except for epochs.

Table 2: The Cross-entropy loss after fine-tuning with a privacy-preserving approach, according to the portion of rewound weights.

| Approach | 0.1% | 1.0% | 3.0% | 5.0% |
|---|---|---|---|---|
| A2 - train | 1.2268 | 0.8570 | 0.4326 | 0.4619 |
| A2 - test | 1.3797 | 1.2728 | 0.9288 | 0.9610 |
| A3 - train | 0.1502 | 0.3376 | 0.4473 | 0.4815 |
| A3 - test | 0.7720 | 0.7433 | 0.8044 | 0.8330 |
| From scratch - train | 0.8087 | | | |
| From scratch - test | 1.5398 | | | |

As for privacy-related information, Tab. 2 displays the model's prediction loss distributions on train and test set at various configurations. It exhibits that CWRF (A3) shows significantly better loss gap compared to A2 and the model trained from scratch, especially at portions of 3.0% & 5.0% while they are at the same testing accuracy at these ratios. Overall, it tells us that fine-tuning on privacy-invulnerable weights (A3) has less negative impact on the testing distribution compared with A2 (fine-tuning on privacy-vulnerable weights.)

## 5 EMPIRICAL STUDY

### 5.1 EXPERIMENTAL SETUPS

**Datasets.** We evaluate defense approaches on three datasets: CIFAR-10 & -100 Krizhevsky et al. (2009) and CINIC-10 Darlow et al. (2018). CINIC-10 contains $270,000$ images, evenly distributed into training, validation, and testing subsets. The size of the images in the CINIC-10 is resized to $32 \times 32$, which is the same as the CIFAR datasets. In all three datasets, we randomly sampled some data points from the training data, which are disjoined from the data points used for training the specific single model. More details regarding sampling are described in MIAs' setting in Appendix B.

**Models.** To adequately evaluate our approach against compared approaches, two commonly used architectures, ResNet18 He et al. (2016) and Vision Transformer (ViT) Dosovitskiy et al. (2021), are used in the experiments. When evaluating with ResNet18, we adapt the model configurations designed for the CIFAR datasets in the original paper. As for ViT, the inputs of images are divided into patches of $4 \times 4$, which is smaller than the ViT designed for the ImageNet dataset Deng et al. (2009) in the original paper.

**Attacks.** To show the superiority of our approach in boosting privacy-preserving methods against membership inference attacks, two recent MIAs techniques, Likelihood Ratio Attack (`LiRA`) Carlini et al. (2022a) and Robust Membership Inference Attack (`RMIA`) Zarifzadeh et al. (2024), are adopted in our defense evaluation. In addition, the strategy of adaptive attacks Song & Mittal (2021) is applied to all MIAs to rigorously evaluate the defense approaches. We evaluate the model's reliance ability against attacks along two metrics: (*i*) *AUC* and (*ii*) *TPR at low FPR*. Specifically, the TPRs at $10^{-3}$ and $10^{-5}$ FPRs are reported in our paper. More details of attacks are elaborated in Appendix B.

**Defenses.** To verify the universality of our approach, we provide extensive comparisons with four privacy-preserving training approaches: Differentially private stochastic gradient descent (`DP-SGD`) Abadi et al. (2016), relaxed loss (`RelaxLoss`) Chen et al. (2022), High accuracy and membership privacy (`HAMP`) Chen & Pattabiraman (2024), convex-concave loss (`CCL`) Liu et al. (2024), and privacy-aware sparsity tuning (`PAST`) Hu et al. (2024) are deployed to train the models against MIAs. We adopt the implementation of DP-SGD provided by the Opacus library Yousefpour et al. (2022) while we adopt the official implementation of other defense approaches. Due to compatibility issues between DP-SGD, Batch Normalization, and Dropout techniques, DP-SGD is only applied to ViT. In addition, since we compare the model's internal privacy-defense ability, the training part of HAMP is deployed when we use it.

**General Configurations.** Adam optimizer Kingma & Ba (2015) is applied to train all models. We set the hyper-parameters $\beta_1 = 0.9, \beta_2 = 0.999$ and the weight decay to $5 \times 10^{-4}$. For the learning rate, we train the model by setting the initial learning rate to $1 \times 10^{-3}$ and changing the learning rate along steps with the cosine annealing scheduler Loshchilov & Hutter (2017). The batch size and epochs of all tasks training from scratch are set to 256 and 100, respectively. As for defenses, we follow the original paper's hyperparameter settings for each approach that we compare with. As for attacks, eight shadow models, including four 'IN' models and four 'OUT' models that are required by LiRA and RMIA, are deployed for both attacks. We report all results in three independent runs. As for the experimental environment, some important information of the computation device is listed as follows:

| CPU | GPU | RAM | OS | CUDA | Python | PyTorch |
|---|---|---|---|---|---|---|
| AMD Ryzen™ 7 7700X | NVIDIA GeForce RTX 5090 | 64 GB | Ubuntu 24.04 LTS | 12.9 | 3.12.3 | 2.80 |

**Customized Configurations** In our approach, on the privacy vulnerability estimation stage, 30 iterations and 256 mini-batch size are applied. The $\lambda$ is set to 0.7 for CIFAR-10 and CINIC-10 while it is 0.9 for CIFAR-100. As for fine-tuning epochs, we set it to 40 with the same initial learning rate using in training from scratch. The same learning rate scheduler is also applied. We perform grid search to select the rewinding rate $r \in [1\%, 10\%]$ in all experiments shown in the main context.

## 5.2 CWRF (Ours) with Various Privacy-Preserving Approaches

In CIFAR-10, we report results with both ResNet18 and ViT in Tab. 3. In the evaluation of ResNet18, three approaches, RelaxLoss, HAMP and CCL are all effective in privacy-preservation. The results exhibit that our approach successfully improves the models' resilience against SOTA MIAs by plugging other privacy-training approaches. Especially, approaches with CWRF all achieve significant mitigation of privacy risks under LiRA. However, under RMIA, the combo of RelaxLoss and CWRF suffers from some slight increase in privacy risks. This is to some extent due to the instability of solely deploying RelaxLoss—the significantly higher variance of test accuracy. With such instability, the shadow models of RMIA become harder to model the target model's behavior. As for ViT, the performance of CWRF becomes even better: combining with all four approaches—DP-SGD, RelaxLoss, HAMP, and CCL, CWRF shows most effective improvements in reliance against the attacks while, in some instances, the testing accuracy becomes even better (DP-SGD + CWRF).

Table 3: The performance of four privacy-preservation approaches with and without CWRF (Ours) on CIFAR-10. Higher is better in test accuracy ($\uparrow$) while lower is better in Privacy ($\downarrow$).

| Model | Defense | Test Acc. (%, $\uparrow$) | LiRA ($\downarrow$) | | | RMIA($\downarrow$) | | |
|---|---|---|---|---|---|---|---|---|
| | | | AUC (%) | TPR(%)@FPR | | AUC(%) | TPR(%)@FPR | |
| | | | | 0.1% | 0.1‰ | | 0.1% | 0.1‰ |
| ResNet18 | No Defense | $79.44_{(0.23)}$ | $85.00_{(2.20)}$ | $2.18_{(0.59)}$ | $1.78_{(0.34)}$ | $74.76_{(1.59)}$ | $5.88_{(0.70)}$ | $3.90_{(1.31)}$ |
| | RelaxLoss | $77.10_{(1.21)}$ | $70.51_{(2.72)}$ | $1.38_{(0.42)}$ | $0.52_{(0.21)}$ | $66.60_{(1.67)}$ | $0.52_{(0.34)}$ | $0.12_{(0.16)}$ |
| | + CWRF (Ours) | $76.86_{(0.29)}$ | $68.31_{(0.68)}$ | $0.03_{(0.05)}$ | $0.03_{(0.05)}$ | $68.18_{(1.53)}$ | $1.22_{(0.97)}$ | $0.27_{(0.19)}$ |
| | HAMP | $77.79_{(0.33)}$ | $79.71_{(0.20)}$ | $3.33_{(0.73)}$ | $1.80_{(1.47)}$ | $80.07_{(0.58)}$ | $7.28_{(1.64)}$ | $1.93_{(1.28)}$ |
| | + CWRF (Ours) | $81.43_{(0.15)}$ | $77.96_{(0.13)}$ | $0.53_{(0.58)}$ | $0.07_{(0.06)}$ | $80.26_{(0.41)}$ | $4.30_{(1.33)}$ | $1.66_{(0.65)}$ |
| | CCL | $79.56_{(0.38)}$ | $83.95_{(0.36)}$ | $1.50_{(0.71)}$ | $0.80_{(0.61)}$ | $76.04_{(0.39)}$ | $4.23_{(0.54)}$ | $2.22_{(1.55)}$ |
| | + CWRF (Ours) | $77.77_{(0.56)}$ | $64.82_{(0.32)}$ | $0.22_{(0.06)}$ | $0.10_{(0.04)}$ | $74.25_{(0.36)}$ | $2.80_{(0.43)}$ | $0.93_{(0.33)}$ |
| ViT | No Defense | $56.45_{(0.46)}$ | $82.88_{(0.68)}$ | $1.60_{(1.14)}$ | $1.92_{(0.41)}$ | $84.44_{(0.27)}$ | $1.52_{(0.81)}$ | $0.45_{(0.32)}$ |
| | DP-SGD | $57.63_{(0.29)}$ | $54.97_{(0.41)}$ | $0.45_{(0.11)}$ | $0.17_{(0.06)}$ | $60.86_{(0.18)}$ | $0.23_{(0.16)}$ | $0.18_{(0.06)}$ |
| | + CWRF (Ours) | $60.45_{(0.37)}$ | $55.68_{(0.58)}$ | $0.13_{(0.06)}$ | $0.00_{(0.00)}$ | $60.46_{(1.03)}$ | $0.13_{(0.02)}$ | $0.03_{(0.05)}$ |
| | RelaxLoss | $57.21_{(0.75)}$ | $73.45_{(0.73)}$ | $0.38_{(0.18)}$ | $0.37_{(0.18)}$ | $72.87_{(1.35)}$ | $0.85_{(0.72)}$ | $0.23_{(0.23)}$ |
| | + CWRF (Ours) | $56.82_{(0.15)}$ | $55.88_{(0.54)}$ | $0.12_{(0.10)}$ | $0.03_{(0.05)}$ | $63.30_{(0.77)}$ | $0.38_{(0.31)}$ | $0.10_{(0.11)}$ |
| | HAMP | $51.62_{(0.72)}$ | $50.53_{(0.41)}$ | $0.07_{(0.09)}$ | $0.00_{(0.00)}$ | $54.42_{(0.55)}$ | $0.27_{(0.12)}$ | $0.05_{(0.04)}$ |
| | + CWRF (Ours) | $52.50_{(0.39)}$ | $50.15_{(0.40)}$ | $0.05_{(0.11)}$ | $0.00_{(0.00)}$ | $51.50_{(1.14)}$ | $0.13_{(0.08)}$ | $0.02_{(0.02)}$ |
| | CCL | $54.25_{(0.71)}$ | $52.18_{(0.53)}$ | $0.02_{(0.02)}$ | $0.00_{(0.00)}$ | $56.33_{(0.83)}$ | $0.12_{(0.08)}$ | $0.00_{(0.00)}$ |
| | + CWRF (Ours) | $53.45_{(0.65)}$ | $51.68_{(0.36)}$ | $0.00_{(0.00)}$ | $0.00_{(0.00)}$ | $51.32_{(0.57)}$ | $0.07_{(0.06)}$ | $0.00_{(0.00)}$ |
| | PAST | $54.84_{(0.56)}$ | $54.30_{(0.79)}$ | $0.17_{(0.10)}$ | $0.08_{(0.08)}$ | $62.99_{(1.42)}$ | $0.97_{(0.25)}$ | $0.25_{(0.25)}$ |
| | + CWRF (Ours) | $54.66_{(0.37)}$ | $53.86_{(1.29)}$ | $0.15_{(0.19)}$ | $0.08_{(0.08)}$ | $62.10_{(0.08)}$ | $0.68_{(0.31)}$ | $0.22_{(0.14)}$ |

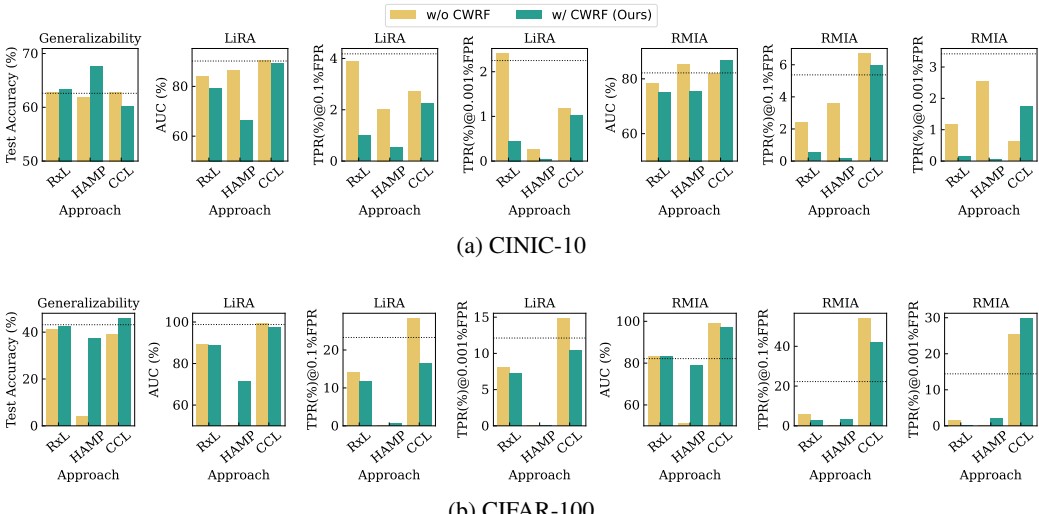

(a) CINIC-10

(b) CIFAR-100

Figure 6: The performance of ResNet18 trained with three privacy-preservation approaches with and without CWRF (Ours). The dotted line represents a baseline performance of a model trained from scratch with regular training approach, Cross-Entropy.

CINIC-10 has more data points, thus showing more stable trends (see Fig. 6a). Considering the utility-privacy tradeoffs, the best combo is HAMP with CWRF: it shows not only a significant advance in test accuracy—even substantially more than the undefended model—but also best privacy resilience against both attacks. However, the CCL is not fully effective under RMIA, the performance becomes worse in terms of AUC and TPR when FPR is fixed at $0.1\%$. After the addition of CWRF, it becomes further worse in RMIA, while the privacy risks are mitigated under LiRA. In RelaxLoss, training with CWRF helps the model stably improve its generalizability and privacy.

In CIFAR-100, the results—see Fig. 6b—vary a lot due to the more difficult task, but limited training samples. We note that the model solely trained with HAMP fails to converge. In contrast, the model can achieve better utility when it is trained with both HAMP and CWRF. As for CCL, the trend is consistent with that in CINIC-10. These results explain that our approach can definitely boost the privacy-preserving approaches only when the approaches can be effective against MIAs. As for RelaxLoss with CWRF, it shows stable improvements in both generalizability and privacy. In addition, in the evaluation of LiRA with $128$ shadow models (discussed in Sec. C.1 in the appendix), CWRF shows the consistent advantages by combining each of the three approaches.

In summary, when the applied privacy-preserving approach is effective in the specific situations, our approach, CWRF, can always boost it to achieve better privacy-utility tradeoffs. We also emphasize that our approach can assist the stability of privacy-preserving training by stabilizing testing accuracy variance through multiple independent runs and avoiding model collapse.

## 6 CONCLUSION

We design a method to estimate weight-level privacy vulnerability. By exploring the correlation between privacy vulnerability and learning ability, we explained and showed why general neural network pruning is not effective in eliminating model privacy vulnerabilities in previous studies. Throughout this paper, we found that privacy vulnerability exists in a very small fraction of weights entangled with learnability. We also recognized the importance of weights stems more from their locations rather than their values. Based on those insights, we propose a strategy to mitigate membership privacy risks of the model that rewinds partial privacy-vulnerable weights and freezes the others, and then does privacy-preserving fine-tuning. Through comprehensive experiments, we demonstrate that our strategy achieves a more effective balance between accuracy and privacy than directly applying existing privacy-preserving methods that train from scratch.

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

# A  FURTHER RELATED WORK

## A.1  MEMBERSHIP PRIVACY PRESERVATION METHODS

Prior membership privacy preservation research Fang et al. (2025) mainly focused on data-end and training components. Abadi et al. (2016) attempted to prevent data points from being over-learned via gradient clipping and noise confusion. Nasr et al. (2018) tried to align member and non-member predictions via adversarial learning. Jia et al. (2019) attempted to mitigate privacy breaches by obfuscating prediction probabilities. Kaya et al. (2020) found that the sense of privacy provided by the regularization mechanisms is false. Chen et al. (2022) designed a prediction-distribution-aligning loss function via reducing the generalization gap and increasing the variance of the training loss distribution. Fang & Kim (2024a;b) attempted to mitigate privacy breach by explicitly facilitating representation alignment in latent space. Liu et al. (2024) achieved privacy preservation by embedding a concave term into convex losses, which help the model predictions with high variance in training losses. Fang & Kim (2026) tried to identify and mitigate the layer-wise start of privacy risks. Zhang et al. (2024) determined that components such as attention modules lead ViTs' privacy vulnerability to be significant than CNNs. Carlini et al. (2022b) observed that simply removing the data identifiable by MIAs from the training dataset induces new privacy leakages in the model. Ye et al. (2024) quantified sample-level privacy vulnerabilities via a leave-one-out approach. Li et al. (2024) tried to separately handle privacy-risky data points that are leaked from model. Yuan & Zhang (2022) observed that common accuracy-oriented pruning & fine-tuning techniques cannot eliminate privacy risks in neural networks. Shang et al. (2025) identified privacy-risky samples to mitigate the privacy risks of the model by rotating the phases of destroying memorization and relearning selective samples during the accuracy-oriented iterative pruning. Shejwalkar & Houmansadr (2021); Tang et al. (2022); Yang et al. (2025) facilitated the mitigation of privacy leakage during training by producing privacy-friendly soft labels. Chen & Pattabiraman (2024) attempted to avoid overconfidence in both training and inference stages. Zhao & Zhang (2025) claimed prior data synthesis approaches cannot prevent privacy leakage. However, past studies did not identify where the privacy risks are inside neural networks. In our paper, we locate and analyze weight-level privacy vulnerabilities.

## A.2  MACHINE UNLEARNING

A general goal of machine unlearning (MU) is to get rid of the impacts of some data points. Current MU approaches can be categorized into two types: (*i*) data reorganization and (*ii*) model manipulation. The data reorganization approaches usually modify data or labels to achieve unlearning, such as label obfuscation Graves et al. (2020), data pruning Bourtoule et al. (2021), or data replacement Cao & Yang (2015). As for model manipulation, it mainly consists of two directions: updating the model weights Schelter (2019); Cha et al. (2024); Georgiev et al. (2025), and replacing components Schelter et al. (2021). In our paper, we mainly study the way of updating model weights to explore the weight-level privacy vulnerability in neural networks.

# B  EXPERIMENTAL SETUPS

**Attacks.** To show the superiority of our approach in boosting privacy-preserving methods against membership inference attacks, two recent MIAs techniques, Likelihood Ratio Attack (`LiRA`) Carlini et al. (2022a) and Robust Membership Inference Attack (`RMIA`) Zarifzadeh et al. (2024), are adopted in our defense evaluation. To simulate the scenario where the shadow model technique Shokri et al. (2017); Carlini et al. (2022a) is applied, only a small portion of the data is sampled as training

Table 4: The number of data points sampled from the entire non-testing set.

| Dataset | Training | Reference |
|---|---|---|
| CIFAR-10 | $18,000$ | $2,000$ |
| CIFAR-100 | $18,000$ | $4,000$ |
| CINIC-10 | $25,000$ | $5,000$ |

data and reference data for each model. In our study, we follow LiRA's sampling strategy, while the precise quantities are different. The specific quantities for each dataset are provided in Tab. 4. In addition, the strategy of adaptive attacks Song & Mittal (2021) is applied to all MIAs to rigorously evaluate the defense approaches. We evaluate the model's reliance ability against attacks along two metrics: (*i*) *AUC*: by integrating the ROC curve across all thresholds, the AUC reflects the degree to which the attacker can distinguish the membership of the data points for the target model that is attacked by attacker; (*ii*) *TPR at low FPR*: we also use true-positive rate (`TPR`) at low false-positive rates (`FPR`) as a metric to show the model's privacy vulnerability since Carlini et al. (2022a) state

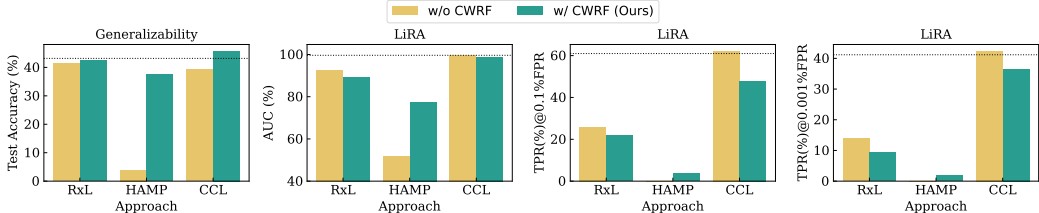

Figure 7: The performance against LiRA when 128 shadow models (64 'IN' and 64 'OUT' models) are deployed for ResNet18 trained with three privacy-preservation approaches (RelaxLoss, HAMP, and CCL) with and without CWRF (Ours) in CIAFR-100. The dotted line represents a baseline performance of a model trained from scratch with regular training approach, Cross-Entropy.

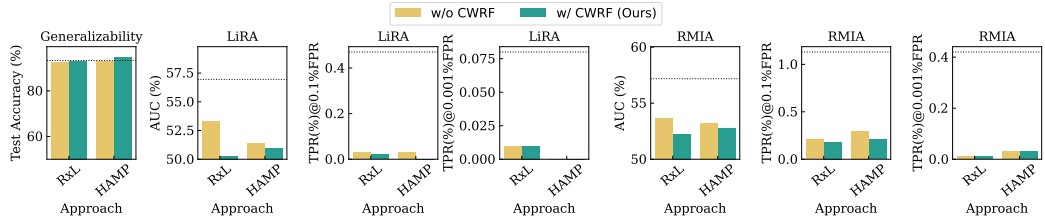

Figure 8: The performance of transformer trained with three privacy-preservation approaches with and without CWRF (Ours) in DBpedia-14. The dotted line represents a baseline performance of a model trained from scratch with regular training approach, Cross-Entropy.

that neither attack accuracy nor AUC scores adequately reflect an attack's ability to confidently determine membership while TPR at low FPR identifies it better. A perfect defense mechanism corresponds to AUC = 0.5 in the first metric while TPR = 0 in the second metric. Specifically, the TPRs at $10^{-3}$ and $10^{-5}$ FPRs are reported in our paper.

## C FURTHER EXPERIMENTAL RESULTS AND DISCUSSION

### C.1 MORE SHADOW MODELS

To reinforce the empirical evidence of our experiments, we further explore how our approach and others perform when evaluate ResNet18 under LiRA with more shadow models in the CIFAR-100 classification task. As shown in Fig. 7, when 128 shadow models, stronger attacks, are deployed, all approaches show more significant privacy flaws, compared with Fig. 6b. Among these approaches, RelaxLoss and CCL show better resisting ability while the utility performance is even slightly better when they are plugged into CWRF, our approach. As for the HAMP, the trends remain the same as Fig. 6b. Through the results, regardless of the number of shadow models, our approach shows consistent advantages when combining with other privacy-training approaches.

### C.2 EVALUATION ON NLP DOMAIN DATASET

To reinforce the empirical evidence of our experiments, we further explore our approach for an NLP dataset — DBpedia-14 Zhang et al. (2015). The DBpedia-14 is an NLP classification dataset that contains 560, 000 training samples and 70, 000 testing samples for fourteen classes from DBpedia. As shown in Fig. 8, we evaluate the approaches with transformer Vaswani et al. (2017). At a similar utility level, combining with CWRF shows improvement in privacy.

### C.3 PRIVACY-UTILITY CURVE

To reinforce the empirical evidence of our experiments, we further explore how our approach and others perform with privacy-utility trade-offs via ResNet18 trained with the CIFAR-100 classification task. As shown in Fig. 9, we show the privacy-utility curve, including the configuration points

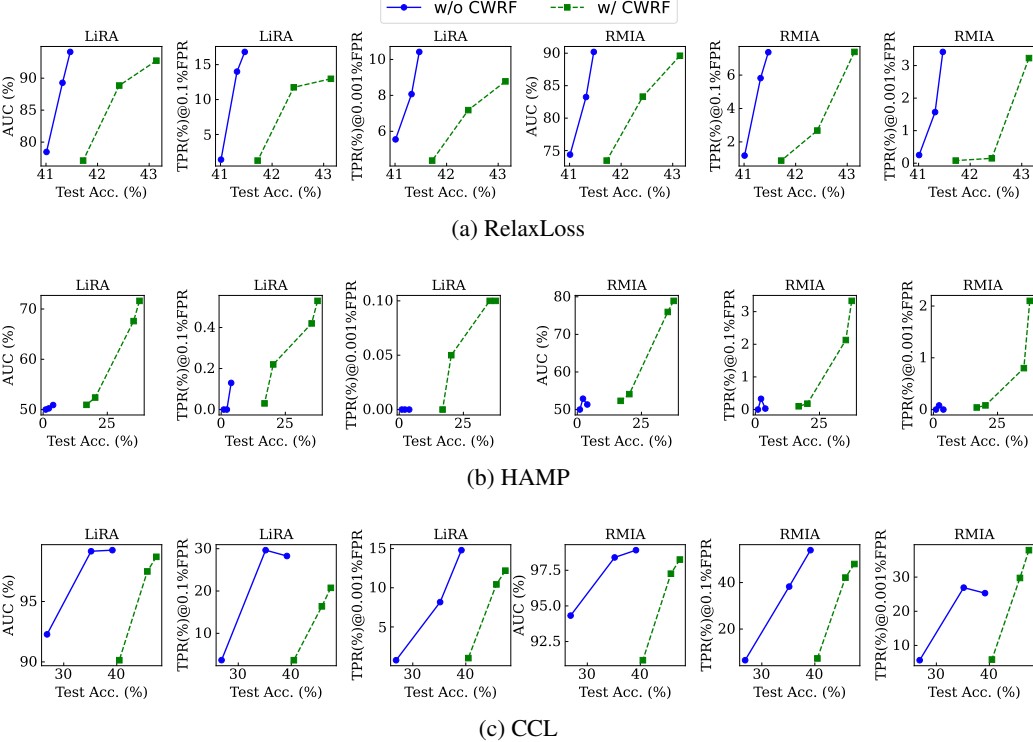

Figure 9: Privacy-utility curve of ResNet18 in CIFAR-100. The bottom right corner (low MIAs yet high test accuracy) is the best performance in terms of privacy-utility.

in Fig. 6b. Compared with the case with each of the three approaches solely, plugging CWRF shows consistent advantages by combining a privacy-training approach.

