# OpenReview forum: "Learnability and Privacy Vulnerability are Entangled in a Few Critical Weights"
_ICLR.cc/2026/Conference — ICLR 2026 Poster_

### Official Review · Reviewer_Jji4 · 2025-10-25

**Soundness:** 2
**Presentation:** 2
**Contribution:** 3
**Rating:** 2
**Confidence:** 4

**Summary:**

This paper finds that performance impact and privacy vulnerability are entangled and exist in a very small fraction of weights. The authors propose a fine-tuning method that curates only privacy-vulnerable weights to defend against membership inference attacks. Experiments show the performance of this method on three datasets.

**Strengths:**

1. The proposed question is interesting. The authors argue that performance impact and privacy vulnerability are entangled and exist in a very small fraction of weights. This is a valuable research problem in the membership inference attack field.

2. The author makes an attempt to confirm the proposed insights by empirical evidence. Based on these observations, this paper proposes a method to mitigate the privacy vulnerability introduced by MIAs.

**Weaknesses:**

1. The presentation of this paper could be improved. A well-organised structure makes it easier for readers to understand the paper and follow the arguments. This paper lacks readability, making it difficult for readers to grasp the viewpoints the author wants to convey. For example, the authors should introduce the importance estimation in neural networks in the preliminaries (not as a related work) if they use the tools as an analytical method.

2. The paper does not establish a clear logical connection between the three proposed insights, the motivation, and the method. The authors claim the importance of weights stems from their locations rather than their values. However, the TFO method estimates weight importance via magnitudes of gradients and weights. Moreover, why does the proposed method not consider fine-tune weights at particular locations, even though the importance of weights stems from their locations? In addition, the authors claim that most privacy-vulnerable weights impact utility performance. Why does rewinding privacy-vulnerable weights, while freezing these weights, have little impact on performance?

3. The experimental results provided don’t sufficiently substantiate the conclusions drawn in this paper. For an empirical paper, authors should conduct extensive experiments to evaluate the effectiveness of the proposed method, including the various attack and defence methods, different types of datasets, and more models. In addition, the authors should conduct experiments to analyse the performance of the method under different configurations, such as hyperparameters, ablation studies on key components. Moreover, the authors should plot privacy-utility curves to show the privacy-utility trade-offs, rather than reporting only a single point result as in Table 3.

**Questions:**

1. In Figure 1(b), does the training loss approach zero, suggesting that the model is severely overfitting? Does CE loss represent privacy vulnerability?

2. How to formally define privacy-vulnerable weight. As described in Section 4, the author uses the weight importance estimation method Eq.2 to estimate privacy vulnerability. Can importance estimation serve as a proxy for privacy vulnerability?
3. Existing paper[1] also propose a membership-privacy-oriented fine-tuning defence method.
4. How to compute the learnability score? Is this the accuracy of models?
5. What does the PCC in Table 1 represent? For example, does a high PCC indicate both high accuracy and a high MIA AUC?
6. Can CWRF be used independently as a defence method?  Why does Table 3 only show the results when CWRF is plugged into other defence methods?
7. The proposed method is similar to pruning techniques. Why are pruning-based defence methods not included in the comparison?

[1] Defending membership inference attacks via privacy-aware sparsity tuning[J]. arXiv preprint arXiv:2410.06814, 2024.

---

> ### Author Response · Authors · 2025-11-20
> **Response - Part 1/4**
>
> #### Thank you for your time and effort in reviewing our manuscript. We have attempted to address all of your questions and concerns. Please let us know if there is anything we missed. Otherwise, we hope you could consider raising your score based on our response. Thank you for your effort again.
>
> **Weakness:**
>
>
> ---
>
> **\[W1\]**
> Thank you for your comment. We kindly ask you to refer to Sec.3 in our original manuscript, where we introduced and described the importance estimation in detail. We discussed it in the **main** section of the paper, because it is one of the important contents of the paper.
>
>
> ---
>
> **\[W2\]**
> We are happy to address all your questions about our methodology:
> **(i) *Why does TFO estimate importance by magnitudes of gradients and weights?***
> TFO was first proposed by the study \[7\]. We must highlight that the **TFO** approximates the importance scores based on **the Leave-One-Out (LOO)** approach. The importance of a parameter is quantified by the error induced by removing it. Under an i.i.d. assumption, this induced error is measured as a squared difference of prediction errors with and without the parameter ($w\_m$): $\\mathcal{I}\_m \= \\bigg({E}(\\mathcal{D},\\textbf{W}) \- {E}(\\mathcal{D},\\textbf{W}|w\_m \= 0)\\bigg)^2$, where $\\mathcal{D}$ denotes the dataset, $\\textbf{W}$ denotes the set of weights/parameters, and $E$ denotes the cost function. In neural networks, $E$ is usually CE loss. Then, the **TFO approximate the error by the First-Order Taylor expansion**, which boils down to a multiplication of gradient and weight magnitude. The detailed derivation process is provided in Sec.3 of the study \[7\] (which is TFO). We will extend this explanation to help readers understand it better. Thank you for your question\!
> **(ii) *Why does the proposed method not consider fine-tune weights at particular locations?***
> *Let’s suppose* two models \- one is naturally privacy-safer than the other while they can achieve similar performance by training. If we apply privacy-preserving approaches to a model to achieve target privacy level (which would hurt utility), it is intuitive that the privacy-safer one’s privacy can be preserved easier with less utility loss. The weights can be understood by this rationale in that training on less privacy-vulnerable weights can help the model sacrifice less utility for privacy if the model can achieve similar utility by merely updating these weights. Hence, there is no benefit to training privacy-vulnerable weights in their locations instead of training privacy-invulnerable weights. Also, a similar utility can be guaranteed by our hypothesis introduced in Sec.4.4. **We highlight that fine-tuning on privacy-vulnerable weights is feasible because the learnability of a weight in a neural network is only determined by its position**.
> **(iii) *Why does rewinding privacy-vulnerable weights, while freezing these weights, have little impact on performance?***
> Thank you for your comment. That is one of the most valuable points we contributed. As introduced by the hypothesis in Sec.4.4, the model’s generalizability stems from the locations, which means it does not have a significant impact whether learnability-critical weights are trained or not. Once we retain these locations of the model, we can still achieve similar performance by training other learnability-unimportant weights. Fig.5 and Tab.2 show the thought process and empirically verify it.
>
>
> ---
>
> **\[W3\]**
> (i) Responding to the reviewer’s comment, we would like to highlight that we did include well-known defense techniques: (1) differentially private stochastic gradient descent (DP-SGD); (2) relaxed loss (RelaxLoss); (3) high accuracy and membership privacy (HAMP); (4) convex-concave loss (CCL). All of them are widely known and recognized privacy-preserving training approaches. In addition, we would like to draw your attention to the fact that these approaches are appreciated by other reviewers as strengths of the submission. (ii) As for attacks, we included two commonly referred SOTA approaches: LiRA and RMIA.

---

> ### Author Response · Authors · 2025-11-20
> **Response - Part 2/4**
>
> **Questions:**
>
> ---
>
> **\[Q1\]**
> (i) We would like to explain the gap between training and testing distributions in Fig.1(b). The gap is more significant than usual when the model is trained with the full train set because we randomly sampled a small portion of the dataset to train the model, which is **required by evaluating on MIAs due to shadow model techniques** as other studies also show \[1,2\]. As for the sampling detail, please kindly refer to Tab.4 and Lines 800-805 in the appendix of the original manuscript. Besides, some prior studies, such as the middle subfigure of Fig.4 in \[1\], the Fig.1(a) in \[3\], the Fig.1(b) in \[4\], and the middle subfigure of Fig.3 in \[5\], also show consistent stats, which is also due to employing shadow model techniques.
> (ii) Yes. The rationale of MIAs is to exploit behavioral differences of an ML model on training data and non-training data. Many MIAs, such as \[1, 6\], use the CE or its variants as the scoring function for attacks. Significant disparity between train vs. test in CE loss is a sign of severe privacy risk of the model.
>
> ---
>
> **\[Q2\]**
> The importance estimation method that accumulates gradient \* magnitude along with training is derived from a well-known pruning paper \[7\]. In our paper, we refer to it as learnability estimation. Actually, learnability is not only generalizability due to the complexity of the training dynamics, but it also represents how a model can be the same or almost the same as the entirely trained model after pruning and fine-tuning in terms of performance, although earlier researchers were mainly focused on generalizability (accuracy). The core difference between learnability estimation and privacy-vulnerability estimation is the objective functions as illustrated in Fig.2. Learnability estimation’s training goal is to select the weights that can learn from the training data via vanilla CE loss while privacy-vulnerability estimation’s training goal is to select weights that can fit the training data while forgetting generalizability (reference data) so that we can select the weights that easily learn features unhelpful to generalizability. In other words, it can select privacy-vulnerable weights. Sec.4.1 explains this insight and idea in detail.

---

> ### Author Response · Authors · 2025-11-20
> **Response - Part 3/4**
>
> **\[Q3\]**
> Thank you for your question. We reviewed the literature that you mentioned and we are happy to discuss it. Prior to paper \[8\] (which you referred to as \[1\] in your review), there was also some discussion \[9\] about the L1/L2 regularization’s impact on privacy. Comparing L1 and L2 Regularization, L1 encourages weights to be zero to suppress weights that do not contribute to optimizing the objective function, while L2 enforces more weights to be close to zero (but not zero) and prevents a small portion of weights from becoming “super impactful” in model decision.  In PAST, they used L1\*Privacy\_Score as the regularization term which is actually similar to L2 but better than L2 since the gradients in \[8\] are based on privacy scores instead of weight magnitude. In this way, it can suppress privacy-risky weights. Among privacy-risky weights, the learnability-unimportant weights would be suppressed prior to the learnability-important ones because the learnability-important weights contribute to CE loss while the unimportant weights do not. Overall, we believe it is good work to further improve the pretrained model’s privacy. **However, there are several superior distinctions of our study from \[8\] as follows:**
> **(i)** First, from the **theoretical** view, one of the significant points that differentiates ours from \[8\] is how we handle privacy-vulnerable yet learnability-critical weights. With our validated hypothesis in Sec.4.4 & Fig.5 (A2 vs. A3), it can be inferred that privacy training on privacy-vulnerable yet learnability-critical weights can lead to damage to the model’s utility.  CWRF is proposed to avoid such damages. However, \[8\] would encourage the model to change the value of these weights by deactivating privacy-vulnerable but learnability-unimportant weights, leading to utility loss. From this aspect, we think CWRF could even work with \[8\] together, possibly with careful codesign, to help \[8\] solve the issue brought by privacy-vulnerable but learnability-weights.
> **(ii)** For the second, as for the **training** paradigm, \[8\] is fine-tuning the model after pretraining with privacy consideration. A significant issue with it is the CE loss. CE loss inherently introduces privacy risks to privacy-vulnerable weights, which can be seen in Fig.1 that prediction disparities are reintroduced by fine-tuning the weights where learnability and privacy-vulnerability are entangled. Besides, fine-tuning with L1 regularization (that \[8\] employed) can be regarded as a type of iterative pruning to some extent. However, \[10\] claimed that iterative magnitude pruning is ineffective in privacy preservation. This could make it difficult to further optimize the balance between privacy and utility. In contrast, we proposed privacy-preserving training by rewinding and freezing the privacy-vulnerable and learnability-important weights.
> **(iii)**  Lastly, we would like to underscore that \[8\] is **not** validated against modern attacks such as, LiRA and RMIA, which are widely recognized as the most powerful empirical proof of membership privacy approaches.  Besides, the most commonly convincing metrics of MIAs such as AUC and TPR@(Low FPR) are also not evaluated in \[8\], although our paper includes all of them.
> Therefore, these factors make our study much more comprehensive and convincing. We will refer to \[8\] in our paper and make these points clear. Also, we included **new results for \[8\]** in Tab.3 of the updated manuscript.
>
> ---
>
> **\[Q4\]**
> We are happy to explain it. The learnability score is not computed with accuracy. Also. learnability does not only mean generalizability, but also represents how a model can be the same or almost the same as the full trained model.  Please kindly refer to Sec.3 and Fig.2(a). In Sec.3, we describe in detail how to compute learnability scores, and the overview of the training dynamics during learnability estimation can be referred to Fig.2(a). Also, at lines 136 & 137 in Sec.3 of the original manuscript, we explicitly defined what learnability is and distinguished the difference between privacy-vulnerability and learnability.
>
> ---
>
> **\[Q5\]**
> PCC denotes Pearson Correlation Coefficient as it was written in the caption of Tab.1 (Line 226 in the submitted version.) Please kindly refer to it.

---

> ### Author Response · Authors · 2025-11-20
> **Response - Part 4/4**
>
> **\[Q6\]**
> Thank you for your question. The CWRF contains three stages: (i) scoring privacy vulnerability, (ii) rewinding and freezing privacy-vulnerable weights according to scores, and (iii) fine-tuning the rest of the trainable weights with a privacy-preserving approach. Please kindly refer to the newly added self-contained description, Alg.1, in the updated manuscript. That is, a privacy training approach can be plugged into the 3rd stage of CWRF for post-training. The relationship between CWRF and privacy-training approaches can be understood as similar to the relationship between pruning and fine-tuning approaches. As shown in Fig.3, the privacy vulnerability is a relative and continuous score rather than an absolute binary. In other words, there can still exist somewhat slightly privacy-vulnerable trainable weights even after rewinding and freezing. To address these weights, we recommend adapting privacy-training approaches for the final fine-tuning. In addition, evaluating combinations of CWRF and multiple training approaches supports the hypothesis presented in Sec.4.4, which is one of the most important contributions in our paper.
>
> ---
>
> **\[Q7\]**
> Thank you for your point. We are eager to discuss the pruning’s impact on privacy and explain why general pruning is actually not good for privacy because it is the important motivation of this work. Firstly, as for traditional learnability pruning, a prior study \[11\] has shown that pruning leads to reintroducing privacy risks, and this issue becomes even more severe when the pruning rate is high. In our paper, we discussed it in Line 116-129 (in the original manuscript) of Sec. 3, which is our motivation for estimating privacy vulnerability by showing why existing learnability pruning does not work. Also, in Sec.4, we analyzed the reason why \[11\] found pruning is not effective for privacy. It is because privacy-vulnerability and learnability are entangled, which is a **novel** insight presented in our paper. One more point that we want to draw your attention to is that \[9\] found that L2 regularization can help enhance privacy to some extent, while L1 cannot. That is because L2 regularization prevents a small portion of weights from becoming “super impactful” in model decision, and this process coincidentally mitigates some memorization from privacy-vulnerable weights to other privacy-invulnerable weights, which makes the model privacy-safer. In contrast, the L1 regularization behaves similarly to general pruning techniques. Finally, we **included new pruning-based results for \[8\]** in Tab.3 in the updated manuscript.
>
> ---
>
> **Reference**
>
> \[1\] Carlini, Nicholas, et al. "Membership inference attacks from first principles." IEEE S\&P. 2022\.
> \[2\] Zarifzadeh, Sajjad, et al. "Low-Cost High-Power Membership Inference Attacks." ICML. 2024\.
> \[3\] Chen, Dingfan, et al. "RelaxLoss: Defending Membership Inference Attacks without Losing Utility." ICLR, 2022\.
> \[4\] Li, Hao, et al. "SeqMIA: sequential-metric based membership inference attack." CCS. 2024
> \[5\] Peng, Yuefeng, et al. "Diffence: Fencing Membership Privacy With Diffusion Models." NDSS. 2025
> \[6\] Song, Liwei, and Prateek Mittal. "Systematic evaluation of privacy risks of machine learning models." USENIX Security Symposium. 2021\.
> \[7\] Molchanov, Pavlo, et al. "Importance estimation for neural network pruning." CVPR. 2019\.
> \[8\] Hu, Qiang, et al. "Defending membership inference attacks via privacy-aware sparsity tuning." ArXiv. 2024\.
> \[9\] Kaya, Yigitcan, et al. "On the effectiveness of regularization against membership inference attacks." ArXiv. 2020\.
> \[10\] Jia, Jinghan, et al. "Model sparsity can simplify machine unlearning." NeurIPS. 2023\.
> \[11\] Yuan, Xiaoyong, and Lan Zhang. "Membership inference attacks and defenses in neural network pruning." USENIX Security Symposium. 2022\.

---

> > ### Author Response · Authors · 2025-11-26
> > **Additional Experiments**
> >
> > We wanted to inform you that, as for privacy-utility curve visualization, we have added Sec. C.3 and Fig. 9 in the appendix of the updated manuscript to show it. We hope this can resolve your concern and strengthen our study. Thank you for your time and effort in reviewing our work again.

---

> > > ### Comment · Reviewer_Jji4 · 2025-11-27
> > >
> > > Thank you for the detailed response. The authors have addressed some concerns, and I agree with raising my review score. There are still concerns:
> > >
> > > 1. CWRF is designed to plug into other defence methods. As shown in Table 3, after applying CWRF, the performance of privacy-utility is inconsistent across different defence methods. Test accuracy decreases for some methods, while AUC improves for others. Can you provide the explanation? It is helpful to clarify the scope of application of the method.
> > >
> > > 2. It is important to report the parameter details of the different attack and defence methods used in the experiments, so that future work can fairly compare their performance. Are the CWRF results in Table 3 the best ones achieved after tuning its parameters? Could you provide more details about the implementation of CWRF and the defence methods in the experiments?

---

> > > > ### Author Response · Authors · 2025-12-02
> > > >
> > > > **\[Q1\]** As can be seen in Tab.3, by being plugged by some approaches — HAMP and DP-SGD — CWRF showed improvement in test accuracy while by some approaches — Relaxloss and CCL — it shows slighlty accuracy drops. We speculate that this utility improvement could be due to the better configurations of signs of weights (discussed in \[1\]) to some extent. Some privacy-training approaches could benefit CWRF in this process while some other approaches could not. Besides, privacy-training approaches could fit different rewinding rates since their privacy-utility curve could vary a lot (e.g., Fig.9 in the appendix), which would lead to privacy-utility trade-offs to some extent.
> > > >
> > > > ---
> > > >
> > > > **\[Q2\]**  We use the models after the last training epoch and report the results with three independent runs. They are not currently optimized for the best performance. The hyper-parameter configuration search is described in the Customized Configurations paragraph of Sec.5.1 of the manuscript. We will include more details about it in the final version.
> > > >
> > > > ---
> > > >
> > > > **Reference**
> > > > \[1\] Gadhikar, Advait Harshal, and Rebekka Burkholz. "Masks, Signs, And Learning Rate Rewinding." ICLR. 2024\.

---

### Official Review · Reviewer_wW66 · 2025-10-26

**Soundness:** 3
**Presentation:** 2
**Contribution:** 3
**Rating:** 6
**Confidence:** 4

**Summary:**

This paper proposes a new algorithm, Critical Weights Rewinding and Finetuning (CWRF), for privacy preservation [without significantly affecting the utility of the model] based on the observation that there is a relatively small fraction of weights that cause privacy leakage in models. They also demonstrate that a large fraction of these privacy-vulnerable weights are critical to the model's learnability. CRWF works by keeping the learnability/ privacy-critical weights intact while fine-tuning the other privacy-invulnerable weights. This work incorporates the concept of importance estimation of weights in a neural network, which is largely used in the context of model pruning (Frankle & Carbin [1]).

**Strengths:**

- Unlike previous works, instead of data points, the author(s) in this paper are concerned with addressing model-level privacy vulnerability in neural networks. They propose a new metric for privacy-vulnerability estimation for individual weights in a neural network based on Bourtoule et al.'s [2] work on machine unlearning.
- They provide empirical evidence to support their conjectures: (a) Privacy-critical weights are concentrated in the layers responsible for learnability (Section 4.1), (b) there is a correlation between privacy-critical and learning-critical weights (Section 4.2), and (c) privacy-vulnerable weights are also least updated during training (Section 4.4).
- CWRF is modular as it allows plugging in any privacy-preserving approach during the fine-tuning step, as demonstrated in Table 3 by integration of CWRF with other privacy-preserving methods such as HAMP, DP-SGD and others. Furthermore, in some cases, CWRF not only improves resistance to MIAs but also boosts the utility (measured in terms of the test accuracy) of the models.
- This work also contributes to the field of Machine Unlearning, where the model trainer aims to minimise/ remove the influence of select data points on a trained machine learning model. With CRWF, the author(s) aim to train a model to mitigate weight-level privacy vulnerability without significantly affecting its utility.

**Weaknesses:**

- In CRWF, per my interpretation of the approach, the model is fine-tuned using the same training dataset as the one used to train $M_{up}$ from scratch in the privacy vulnerability estimation step. However, the author(s) do not clarify this in Section 4.3.
The author(s) do not clarify what privacy parameters, for example, $(\epsilon, \delta)$, were used in the experiments with DP-SGD.
- In lines 388-389, the author(s) state that they use a total of 8 shadow models for both LiRA and RMIA. While RMIA is said to work well with a low number of shadow models, comparing it against a powerful version of LiRA would necessitate training more than 8 shadow models. Carlini et al. [3] use 256 shadow models. If the author(s) intent is to convey the strength of their proposed privacy-preservation approach (CWRF), they ought to evaluate it using strong attacks and not its weaker versions.
- Lines 404-405 that "It is verified that our approach successfully boosts the performance of all of them." ought to be framed in the context of the performance of CWRF + other privacy-preserving methods against SOTA MIAs.
- Carlini et al. [4] demonstrated the onion-effect wherein removing privacy-vulnerable data points from training exposes earlier invulnerable data points to replace them. Do the author(s) probe whether something similar happens when using CWRF, which involves rewinding privacy-vulnerable weights to their initial values? Could this cause previously privacy-invulnerable weights to become privacy-critical?
- The author(s) do not share the code for the paper.

[1] Frankle, J., and Carbin, M. The Lottery Ticket Hypothesis: Finding Sparse, Trainable Neural Networks. ICLR 2019.

[2] Bourtoule, L. et al. Machine Unlearning.” IEEE S&P 2021.

[3] Carlini, N., et al. Membership Inference Attacks From First Principles. IEEE S&P 2022.

[4] Carlini, N. et al. The Privacy Onion Effect: Memorization is Relative. NeurIPS 2022.

**Questions:**

**Questions**: I would urge the author(s) to address the weaknesses detailed above.

**Suggestions**:

Author(s) can improve the presentation of the paper by considering the following suggested edits:

- Minor suggestion #1: It might be good to rearrange the Fig 3 and Tab 1's results by decreasing order of PCC for an easy read.
- Minor suggestion #2: The PCC reported for linear layer weights for ResNet18 in Tab 1 does not appear aligned with the results as depicted in 1st row, 4th column of Fig 3. If possible, use a different y-axis to demonstrate the high correlation between privacy vulnerability and learnability for linear layers (~0.8).
- Minor Suggestion #3: I would urge the authors to use the full form of the abbreviated terms, such as HAMP and CCL, in Line 377, when referencing them for the first time. Thereafter, they can use the abbreviation. It would be erroneous to presume a reader knows the abbreviations beforehand.
- Minor Correction #1: Line 811, Specifically <-> "Specificly".

---

> ### Author Response · Authors · 2025-11-22
> **Response - Part 1/2**
>
> Thank you for your effort and time in reviewing our manuscript. We also appreciate your patience. It took some time to get the results that you asked for. Hope that would help resolve your concerns.
>
> ---
>
> **For Weaknesses:**
>
> ---
>
> **\[W1\]** Thank you for your comment. As for the hyper-parameters tuning in DP-SGD, we tune the clipping bound in the interval \[1.0, 4.0\]  when searching for the noise scale in {0.01, 0.05, 0.1} for the three datasets, as suggested by the official documents of Opacus library \[1\] and other published studies \[2,3\]. We will add a description about hyper-parameter searching and settings of all approaches. Please let us know if you have any further suggestions regarding this matter. We will follow up.
>
> ---
>
> **\[W2\]** Thank you for your valuable suggestion. In our evaluation, we included an adaptive attack policy \[5\] in both LiRA and RMIA, which means that the shadow models are the same as the target models in terms of architecture and training configuration. As prior studies showed the performance boost of MIAs through an adaptive attack \[5\], such as Fig.12 in the LiRA paper \[4\] and Tab.2 in the RelaxLoss paper \[2\], we believe it is a strong indicator to show our CWRF’s effectiveness. Having said that, we analyzed the models trained without defense under LiRA attacks with more shadow models in CIFAR-100. As shown in the table below:
>
> | \# Shadow Models | AUC (%) | TPR (%) @ 0.1% FPR | TPR (%) @ 0.1‱ FPR |
> | :----: | :----: | :----: | :----: |
> | 8 | 90.19 | 4.18 | 2.25 |
> | 64 | 98.79 | 23.30 | 12.12 |
> | 128 | 99.48 | 51.95 | 41.93 |
> | 256 | 99.62 | 60.98 | 41.77 |
>
> The table shows the results of LiRA in ResNet-18 trained without defense. With this table, we absolutely agree with your suggestion that more shadow models can show clearer trends. Hence, considering a reasonable time and resource cost and the result that the TPR under 0.1‱ changes insignificantly when the number of shadow models is doubled from 128 to 256, we show the results with 128 shadow models to reinforce CWRF’s effectiveness. Please kindly refer to Fig.7 and Sec.C in the appendix and the updated texts in the 10th page of the updated main paper. Under the LiRA with 128 shadow models, our approach shows the consistent trend to the case with the LiRA with 8 shadow models (Fig.6b in the original manuscript).
>
> ---
>
>
> **\[W3\]** Thank you for your suggestion. Per your suggestion, we rephrased the corresponding sentence. Please kindly refer to Lines 446-448 in the revised manuscript.
>
> ---
>
> **\[W4\]** It is absolutely an interesting point. From a high-level view, memorization in a weight can be migrated to other weights. This has been widely validated by many pruning papers, especially structural pruning (e.g., \[6\]) that a model can recover from a model crash (in terms of learning capability) (if pruning does not affect memorization, the model should not crash) via fine-tuning after pruning. Besides, Fig.4 in our manuscript also shows that the lost accuracy (M3) is recovered (M2) by fine-tuning. Having said that, it does not mean all kinds of memorization can be transferred. In Fig.5, with rewinding and freezing 0.1% of most privacy-vulnerable weights, A3 can produce even better performance in a few epochs of fine-tuning. Also the performance becomes closer to the original model’s when the rewinding rate is increased. This phenomenon suggests that there is indeed behavioral inconsistency in the model when different weights are involved in training. In other words, something similar to onion-effect may happen since the privacy vulnerability scores are continuous rather than  binary. This support  why we propose plugging privacy-training approaches into the CWRF. At the same time, we also highlight that memorization cannot be fully migrated since the performance of A3 shows a clear trend (the training loss goes up along with the increase of rewinding rate) when fine-tuning with various portions of weights but with the same training configuration in Tab.2 of the manuscript.
>
> ---
>
> **\[W5\]** We will open-source our code upon acceptance of this paper. We just did not upon submission, just because the ICLR submissions, including review processes, are fully open to the public.

---

> ### Author Response · Authors · 2025-11-22
> **Response - Part 2/2**
>
> **For Suggestions:**
>
> ---
>
> **\[S1\]** Yes, we agree with your suggestion \- it definitely makes them easier to read.  We rearranged them according to your suggestion. Please kindly check Fig.3 and Tab.1 in the updated manuscript.
>
> ---
>
> **\[S2\]** Thank you for your comment. Right, as you mentioned, the aligned range does not effectively show a clear trend between learnability and privacy vulnerability scores. As can be seen in the 2nd row, 4th column of Fig 3 (the linear layers in ViT) in the original manuscript, the correlations can be shown better in adaptive scale range. However, the reviewer **`fnse`** suggested aligning the range. Since the two suggestions are conflicting, we will follow either decision after a discussion among the reviewers and the AC. In the meantime, Tab.1 should help supplement Fig. 3 by showing quantified correlations between learnability and privacy vulnerability scores. We believe that Tab.1 is a helpful interpretation and supplement to Fig.3. Please let us know if you have any further suggestions about it. We will follow up.
>
> ---
>
> **\[S3\]** Thank you for your suggestion. We added the full form of the abbreviated terms in the description of experimental setups to improve our paper’s readability. Please kindly refer to the “Defenses” paragraph in Sec.5.2 on Page 8 to see if they read well.
>
> ---
>
> **\[S4\]** Thank you for pointing it out. We corrected the word in our updated manuscript.
>
>
> ---
> **Reference**
>
> \[1\] Opacus library: [https://github.com/pytorch/opacus](https://github.com/pytorch/opacus)
> \[2\] Chen, Dingfan, et al. "RelaxLoss: Defending Membership Inference Attacks without Losing Utility." ICLR, 2022\.
> \[3\] Chen, Zitao, et al. "Overconfidence is a dangerous thing: Mitigating membership inference attacks by enforcing less confident prediction." NDSS, 2024
> \[4\] Carlini, Nicholas, et al. "Membership inference attacks from first principles." IEEE S\&P. 2022\.
> \[5\] Song, Liwei, et al. "Systematic evaluation of privacy risks of machine learning models." USENIX Security. 2021\.
> \[6\] Molchanov, Pavlo, et al. "Importance estimation for neural network pruning." CVPR. 2019\.

---

> ### Comment · Reviewer_wW66 · 2025-11-24
> **Official Comment to Authors' Rebuttal**
>
> Thanks for your detailed response. The changes made to the paper do strengthen its contributions. I will be updating my initial score to recommend an Accept. That being said, I have one minor suggestion: The presentation of Algorithm 1 (Page 6) could use some editing. The grey colour used for comments is hard to read. Do consider using a dark colour palette for them.

---

> > ### Author Response · Authors · 2025-11-24
> >
> > Thank you for your recognition of the value of our work. As per your suggestion, we updated Algorithm 1. Hope this looks visually easier. Please let us know if there is anything further to improve. We again appreciate that you participated and involved in the discussion actively and timely, and that you improved the quality of our work.

---

### Official Review · Reviewer_XTZA · 2025-10-29

**Soundness:** 4
**Presentation:** 2
**Contribution:** 3
**Rating:** 8
**Confidence:** 4

**Summary:**

The paper provides evidence that privacy vulnerability (as quantified by susceptibility to membership inference attacks, or MIAs) is concentrated in certain locations/weights of a model. Additionally, these weights are also those that correspond to learnability/generalizability. With these insights, the authors propose a method that rewinds and freezes these privacy-vulnerable weights and fine-tunes the other unfrozen weights with (potentially private) information with privacy-preserving approaches such as DP-SGD and more recent approaches. They further corroborate their claims with an extensive array of results across different percentages of rewound weights, different methods, ablating which weights are rewound/fine-tuned, etc. along with an analysis of privacy-utility tradeoff for their method and the reduction in attack success observed for different state-of-the-art MIAs (LiRA and RMIA).

**Strengths:**

**[S1]** Provides valuable insights on which weights correspond to MIA vulnerability and how they are also largely the same as the ones that are most important to generalizability. The methods used to identify such weights are sound and convincing.

**[S2]** Insights provided in Fig. 5 to motivate the design of CWRF are interesting; it clearly demonstrates why it is key to both rewind privacy vulnerable weights and fine-tune privacy-invulnerable (so to speak) weights. It is also interesting how fine-tuning privacy-invulnerable weights, even if they might not be most correlated with learnability, helps significantly with achieving a good privacy-utility tradeoff/test accuracy. In other terms, the observation that the location of these weights, and not their values, is what matters is fascinating, albeit not very intuitive.

**[S3]** Demonstrates the effectiveness of the proposed method (CWRF) when paired with multiple prominent differentially private training algorithms.

**[S4]** Uses the strongest possible attacks to test their proposed defense against. In addition, they report attack success in low FPR regions, which is absolutely essential to effectively communicate privacy attack success rates (as argued by [1]).

**[S5]** In-depth descriptions of the hyperparameters, libraries/software, and hardware used are provided, providing confidence in the reproducibility of the work.

All in all, this paper appears to provide a valuable contribution to the community: a method that can reliably defend against powerful MIAs while maintaining a good level of test accuracy/utility. I find the contributions of this paper valuable and appealing.

---

## References:

**[1]** Carlini, Nicholas et al. “Membership Inference Attacks From First Principles.” 2022 IEEE Symposium on Security and Privacy (SP) (2021): 1897-1914.

**Weaknesses:**

**[W1]** The description of the method could be done more clearly. While the motivation of the methodology by discussing prior approaches to rewinding, weights freezing and fine-tuning, and ablation studies is a great choice, it will be highly beneficial to have a concise self-contained discussion about the steps involved in CWRF along with pseudocode to avoid ambiguity and for easy reference.

**[W2]** The text in tables 1 and 2 is very small and not very appropriate for a potential camera-ready version.

---

This paper is pretty solid otherwise.

**Questions:**

**[Q1]** Can you please address W1 and add a self-contained description of CWRF in the final version of the paper for better presentation?

**[Q2]** Could you also ideally address W2 as well?

---

> ### Author Response · Authors · 2025-11-20
>
> **For Questions:**
>
> \
> **\[Q1\]** Thank you for your valuable suggestion.
> (1) The CWRF contains three stages: (i) scoring privacy vulnerability, (ii) rewinding and freezing privacy-vulnerable weights according to scores, and (iii) fine-tuning the rest of the trainable weights with a privacy-preserving approach. CWRF can adapt arbitrary privacy training approaches by plugging them into the 3rd stage of CWRF for post-training.
> (2) Thanks to your suggestion, we made a self-contained description of CWRF by adding a pseudo-code algorithm. Please kindly refer to Alg.1, which is included in Sec. C of the appendix in the updated manuscript.
>
> **\[Q2\]** Thank you for reviewing our paper in detail. Thanks to your comment, we increased the font size of Tab.1 & 2  to improve their readability. Please kindly check the current Tab.1 & 2 in the updated manuscript. We hope they read better now.

---

> > ### Comment · Reviewer_XTZA · 2025-11-20
> >
> > Thank you for addressing my concerns and for the additions! The figures look better already and the discussion of CWRF is better presented now. I appreciate the text you have added in the rebuttal; it'd be ideal if you could add this description in the main paper where you reference Appendix C for a better explanation.
> >
> > If it helps, ICLR allows an additional page during the rebuttal phase/for the camera-ready version, so I believe you have ample space for a proper discussion/description, and I'd encourage the authors to include that.

---

> > > ### Author Response · Authors · 2025-11-20
> > >
> > > Thank you for your valuable suggestion. As per your suggestion, we moved the algorithm into the main paper and also added a description for the workflow of CWRF in the main text. Please check if they read well in Line 316-320 in the updated manuscript. Please let us know if there are any other concerns or questions you may have. Otherwise, we sincerely hope you could consider raising your score. Thank you.

---

> > > > ### Comment · Reviewer_XTZA · 2025-11-20
> > > >
> > > > I appreciate the authors making this change! I believe it is a step in the right direction for readability. I suggested these cosmetic changes for the sake of better presentation for a venue like ICLR but also to help readers better navigate and appreciate the existing scientific results in the initial submission.
> > > >
> > > > That said, my current score already reflects strong and firm support for the paper's acceptance, and I may need to engage in discussion with fellow reviewers (who have a number of concerns of their own) to ascertain whether I'd go for a perfect 10 for an award-worthy score.
> > > >
> > > > That said, this paper has some neat insights the community can benefit from, and I'll make sure to champion this paper during future AC-reviewer discussions.

---

> > > > > ### Author Response · Authors · 2025-11-20
> > > > >
> > > > > Thank you for your prompt and close engagement in the discussion, as well as your recognition of the value of our paper. We greatly appreciate it!

---

> > > > > > ### Comment · Reviewer_XTZA · 2025-11-20
> > > > > >
> > > > > > I appreciate the enthusiastic engagement from the authors and their hard work/scientific contribution. I personally enjoyed reading your work, and I wish you all the best with this submission!

---

> > > > > > > ### Author Response · Authors · 2025-11-20
> > > > > > >
> > > > > > > Thank you very much!

---

### Official Review · Reviewer_fnse · 2025-10-30

**Soundness:** 3
**Presentation:** 3
**Contribution:** 2
**Rating:** 4
**Confidence:** 5

**Summary:**

This paper introduces a method, Critical Weights Rewinding and Finetuning (CWRF), for optimizing model utility and membership privacy simultaneously. It attempts to resolve the issue of "entanglement", where learnability (utility) and privacy vulnerability are concentrated in the same small set of critical weights. The new approach is applied as a "booster" to existing privacy-preserving training methods (DP-SGD, RelaxLoss, HAMP, and CCL) and is evaluated on both model utility and its resilience to Membership Inference Attacks (MIAs). The algorithmic contribution is demonstrated with a suite of experiments in classic benchmark domains (CIFAR-10, CIFAR-100, and CINIC-10) and on modern architectures (ResNet18 and ViT).

**Strengths:**

S1. The paper is well written and easy to follow

S2. The insights are interesting and helpful

**Weaknesses:**

I found the claims in the paper are interesting but unconvincing due to several points:

W1. Lacks of theoretical motivation. The paper's central hypothesis that a weight's importance stems from its location, not its value is a strong one, but it is presented without theoretical proof and is supported only by a specific set of ablation studies. The paper argues that because A3 (CWRF) successfully recovers accuracy while A1 (Remove) fails, the hypothesis is validated. While the result is compelling, it is not a proof. Can the authors provide any formal argument for this? How does this "location-over-value" hypothesis relate to existing work like the Lottery Ticket Hypothesis, which is mentioned but not deeply connected?

W2. Missing datasets. Previous baselines such as RelaxLoss or HAMP include tabular dataset such as Purchase or Texas besides image dataset.

W3. The justification for fine-tuning invulnerable weights (A3) over vulnerable weights (A2) is the paper's most critical experimental result. However, this experiment appears to have been run only using RelaxLoss as the fine-tuning defense. Does this crucial finding hold for DP-SGD, HAMP, and CCL as well?

**Questions:**

Please refer to the Weaknesses section.

Some minor points:

- The abstract states CWRF "exhibits outperforming resilience", but the actual results show mixed performance on RMIA for ResNet18 e.g RelaxLoss is better without CWRF

- In Figure 3, the paper notes that the axes for the ViT plots are not consistent. While this is acknowledged, it makes visual comparison of the correlations across different layer types (e.g., Att+MLP vs. Norm) very difficult.

---

> ### Author Response · Authors · 2025-11-26
> **Response - Part 1/2**
>
> Thank you very much for your patience. It took a lot longer than we expected to obtain additional experimental results. Please find our response to your comments below:
>
> ---
>
> **Weakness:**
>
> ---
>
> **\[W1\]** Thank you for your questions.  You are right that this paper aims to show the idea empirically, which we believe has its own value, like the Lottery Ticket Hypothesis (LTH) paper \[1\] which is a pure empirical paper, but is insightful and impactful. Per your suggestion, we are happy to discuss the relationship between the Lottery Ticket Hypothesis and our hypothesis:
> The Lottery Ticket Hypothesis (LTH) demonstrates that a subnetwork with learnability-critical weights can perform as well as the entire network when they are trained from the same initialization. This hints that both location and magnitude are important for learnability and updating these learnability-critical weights may have a comparable impact to updating all weights of the unpruned network. In our hypothesis, we think the location is the key to the network’s learnability rather than the magnitudes. That is, the “winning” subnetwork was already born when the network’s architecture was formed and before the weights were initialized. Since the magnitudes of weights barely determine the weights’ learnability, updating the value of the learnability-critical weights may not be critically necessary for the model to converge. In Fig. 5, we verified that the model can recover its accuracy without updating privacy-vulnerable weights, which are entangled with learnability-critical weights, while the model becomes untrainable when these weights are absent. Combining the evidence in LTH paper \[1\], we show that a model is trainable as long as it has the proper number of updatable weights at learnability critical locations. We argue that Fig. 5 of the main paper is a clear empirical proof because it strictly abides by the method of controlling variables, which changes only a single independent variable, i.e., the presence or absence of learnability-critical location.
>
> ---
>
> **\[W2\]** Thank you for your comment. We evaluated our approach for an NLP classification dataset, DBpedia-14 \[2\], and on transformer models \[3\]. Please kindly refer to Sec.C.2 and Fig.8 in the appendix of the updated manuscript.
>
> ---
>
> **\[W3\]** Thank you for your question. Yes, the finding also holds for all other privacy-training approaches besides RelaxLoss \[3\]. For instance, we trained HAMP on  DBpedia-14, and the results are shown in the following table:
>
> | Approach | Test Acc (%) |
> | :---: | :---: |
> | CE | 93.33 |
> | HAMP | 92.75 |
> | CWRF (5% rewinding \+ HAMP) | 95.09 |
>
> As can be seen, the results are consistent to Fig.5 of the original paper that CWRF boosts the model utility when fine-tunined with a privacy-training approach (Similar trend can be also seen in Fig.6a and Tab.1—ResNet18 HAMP part—of the original paper). In other words, CWRF also receives privacy benefits while HAMP loses some utility. We speculate that this utility improvement could be due to the better configurations of signs of weights (discussed in  \[5\]) to some extent. Additionally, \[4\] also found that jointly using multiple training techniques could be beneficial to utility, which is also a support for our speculation. Compared to this, it is challenging to achieve such accuracy when deploying privacy-training on a small number of learnability critical weights, which are entangled with privacy-vulnerable weights. Altering these weights tends to have a significant negative impact on accuracy.
>
> In fact, a considerable amount of utility can be recovered in the 1st epoch of fine-tuning when we rewind and freeze the most privacy-vulnerable weights and fine-tune the remaining weights. Moreover, we find some privacy-training approaches cannot successfully achieve better privacy while maintaining high utility, as we see in the case of HAMP in Fig.6 of the main paper and Fig.7 as well as Fig.9b of the appendix in the updated manuscript. In these cases, CWRF can make the model easier to train.

---

> ### Author Response · Authors · 2025-11-26
> **Response - Part 2/2**
>
> **Minor Points:**
>
> ---
>
> **\[P1\]** Thank you for your valuable question, we are happy to explain and further expand it. We Interpret that it could be due to unstable training with RelaxLoss for the case. In ResNet18 training, RelaxLoss shows a higher standard deviation and better privacy-level among the three approaches. However, in ViT, the standard deviation of the three approaches are lower and for the other two approaches, it outperforms. Therefore, high standard deviation when training ResNet18 with RelaxLoss in CIFAR-10 could be a factor that affects the MIAs performance since we deploy the adaptive attack policy \[6\] that trains shadow models using the corresponding techniques and configuration into the two MIAs. Having said that, respecting your comment, we have rephrased it to ensure it is precisely correct. Please kindly check it out.
>
> ---
>
> **\[P2\]** Thank you for your comment. Right, as you mentioned, we acknowledged that the ranges could not be aligned. According to your comment, we have tried to align them for ViT again, but that washed out the detailed trends that we must read from the charts \- because when the range is too wide, all the data points are just clustered in a very small region. This was why we have Tab. 1, since the visualization is not perfectly ideal to show differences between all pairs \- in Tab. 1 we quantified the correlations of learnability score and privacy-vulnerability score in each type of layers via Pearson correlation coefficient. To make Fig.3 clearer, we provide a table to further describe ViT’s Privacy-Vulnerability Scores:
>
> | Weight Type | Min | Max | Mean |
> | :---- | :---- | :---- | :---- |
> | All | 0.0 | 9.39 | 7.23e-4 |
> | Att+MLP | 0.0 | 1.58 | 6.38e-4 |
> | Linear | 4.3e-43 | 0.24 | 6.19e-3 |
> | Norm | 0.0 | 9.39 | 7.97e-2 |
>
> As shown in the table above, the Norm layers have larger portions of privacy-vulnerable weights since all types’ minimum scores are almost the same while Norm layers’ maximum score and mean score are significantly higher than other types of layers (which tells that the Norm layers have a lot longer tail than the others). Combining it with Tab.1 in our manuscript, it is inferred that there are larger portions of privacy-vulnerable but learnability-unimportant weights in the  Norm layers.
>
> ---
>
> **Reference**
> \[1\] Frankle, Jonathan, and Michael Carbin. "The Lottery Ticket Hypothesis: Finding Sparse, Trainable Neural Networks." ICLR. 2019\.
> \[2\] Zhang, Xiang, Junbo Zhao, and Yann LeCun. "Character-level convolutional networks for text classification." NeurIPS. 2015\.
> \[3\] Vaswani, Ashish, et al. "Attention is all you need."  NeurIPS. 2017\.
> \[4\] Chen, Dingfan, et al. "RelaxLoss: Defending Membership Inference Attacks without Losing Utility." ICLR, 2022\.
> \[5\] Gadhikar, Advait Harshal, and Rebekka Burkholz. "Masks, Signs, And Learning Rate Rewinding." ICLR. 2024\.
> \[6\] Song, Liwei, and Prateek Mittal. "Systematic evaluation of privacy risks of machine learning models." USENIX Security Symposium. 2021\.

---

> > ### Comment · Reviewer_fnse · 2025-11-28
> >
> > Thank you for the detailed response. The authors have addressed most of the critical weaknesses that I mentioned in the review. Therefore, I raise my score towards acceptance.
> >
> > Please take into account the suggested changes in the final version.

---

### Official Review · Reviewer_wcwv · 2025-10-31

**Soundness:** 3
**Presentation:** 3
**Contribution:** 2
**Rating:** 6
**Confidence:** 5

**Summary:**

This paper investigates the relationship between model learnability and privacy, finding that privacy vulnerability is "entangled" with utility performance within a very small fraction of critical weights. The authors propose a key insight that the importance of these weights stems from their location rather than their trained values. Based on this, the paper introduces Critical Weights Rewinding and Finetuning (CWRF) , a strategy that identifies this small set of privacy-vulnerable weights. Instead of pruning, CWRF rewinds only these high-risk weights to their initial, privacy-safe values and then freezes them. Finally, the model recovers its utility by fine-tuning only the remaining "privacy-invulnerable" weights, leveraging the fact that the critical weights' locations are preserved. Experiments demonstrate that CWRF successfully boosts the effectiveness of existing privacy-preserving training methods, achieving a superior privacy-utility tradeoff.

**Strengths:**

1. The work offers a significant contribution by clearly explaining why standard model pruning fails to mitigate privacy risks, linking it directly to this entanglement.
2. The paper proposes CWRF that cleverly combines machine unlearning for vulnerability estimation with weight rewinding, which boosts existing privacy-preserving methods to achieve.

**Weaknesses:**

1. The paper provides no sensitivity analysis for the hyperparameter rewinding rate $r$ and $\lambda$, making it unclear how to set it efficiently.
2. The empirical validation is limited to small-scale models (ResNet18, small ViT) and datasets (CIFAR, CINIC). It is not demonstrated whether the vulnerability estimation step is computationally feasible or if the core insight scales to LLMs.
3. It is recommended to include full privacy–utility curves (similar to those reported for RelaxLoss and CCL) rather than only isolated metric points, as these curves may provide a more comprehensive representation of the privacy-utility trade-off.
4. Previous work [1] also discusses which parameters substantially impact privacy risk. Please add a discussion and comparison with this work.

[1] "Defending Membership Inference Attacks via Privacy-aware Sparsity Tuning" (2024)

**Questions:**

1. How are the hyperparameters chosen for the baseline methods reported in this paper? For example, the loss threshold for RelaxLoss and the $\gamma$ for RMIA.
2. How do the authors split the dataset for training the target model and reference models?

---

> ### Author Response · Authors · 2025-11-24
> **Response - Part 1/3**
>
> **Weakness:**
>
> ---
>
> \[W1\]
> Thank you for your comment. Please kindly refer to the “Customized Configurations” paragraph of the Sec.5.1. We also want to draw your attention to that  A3 (CWRF) in Tab. 2 also showed the impact of rewinding rates regarding CE loss changes in member and non-member data. It exhibits that a higher rewinding rate can achieve better privacy without changing other configurations, although it is also possible to lose some utility with the increase of the rewinding rate (shown in Fig.5).
>
> ---
>
> \[W2\]
> Thank you for your comment. As for the estimation step, our approach estimates the vulnerability just in a few tens of iterations (not epochs; in our paper, it is usually 30 iterations), which results in a low computation cost (much less than a vanilla training epoch).
>
> As for LLM tasks, we studied literature regarding current research on membership privacy in LLMs. We realized it would require more careful and deeper exploration, as the positions of the existing literature are mixed. Although we believe it would be feasible to deploy it  in an LLM model, due to institutional resource limitations, we could not show results on LLM models. Please understand that to show such results for MIAs that we showed in the paper, they require experimental setups with several combinations of components to compare, such as defense approaches, privacy training approaches, shadow model trainings, attack deployments, etc. If the reviewer suggests, we will mention that the results are not conducted on LLM models. Please advise. For the last remark, we believe your suggestion will open an important future research avenue for other researchers and also for us.
>
> ---
>
> \[W3\]
> Thank you for your comment. We included new results regarding privacy–utility curves. Please kindly refer to Fig.9, which is included in Sec.C.3 of the appendix in the updated manuscript.

---

> ### Author Response · Authors · 2025-11-24
> **Response - Part 2/3**
>
> \[W4\]
> Thank you for your point. We reviewed the literature that you mentioned and we are happy to discuss it. Prior to paper \[1\] (which you referred to as \[1\]), there was also some discussion \[2\] about the L1/L2 regularization’s impact on privacy. Comparing L1 and L2 Regularization, L1 encourages weights to be zero to suppress weights that do not contribute to optimizing the objective function, while L2 enforces more weights to be close to zero (but not zero) and prevents a small portion of weights from becoming “super impactful” in model decision.  In PAST, they used L1\*Privacy\_Score as the regularization term which is actually similar to L2 but better than L2 since the gradients in \[1\] are based on privacy scores instead of weight magnitude. In this way, it can suppress privacy-risky weights. Among privacy-risky weights, the learnability-unimportant weights would be suppressed prior to the learnability-important ones because the learnability-important weights contribute to CE loss while the unimportant weights do not. Overall, we believe it is a good work to further improve the pretrained model’s privacy. However, there are several superior distinctions of our study from \[1\].
>
> (i) First, from the theoretical view, one of the significant points that differentiates ours from \[1\] is how we handle privacy-vulnerable yet learnability-critical weights. With our validated hypothesis in Sec.4.4 & Fig.5 (A2 v.s. A3), it can be inferred that privacy training on privacy-vulnerable yet learnability-critical weights can lead to damage to the model’s utility.  CWRF is proposed to avoid such damages. However, \[1\] would encourage the model to change the value of these weights by deactivating privacy-vulnerable but learnability-unimportant weights, leading to utility loss. From this aspect, we think CWRF could even work with \[1\] together, possibly with careful codesign, to help \[1\] solve the issue brought by privacy-vulnerable but learnability-weights.
>
> (ii) For the second, as for the training paradigm, \[1\] is fine-tuning the model after pretraining with privacy. A significant issue with it is the CE loss. CE loss inherently introduces privacy risks to privacy-vulnerable weights, which can be seen in Fig.1 that prediction disparities are reintroduced by fine-tuning the weights where learnability and privacy-vulnerability are entangled. Besides, fine-tuning with L1 regularization (that \[1\] employed) can be regarded as a type of iterative pruning to some extent. However, \[3\] claimed that iterative magnitude pruning is ineffective in privacy preservation. This could make it difficult to further optimize the balance between privacy and utility. In contrast, we proposed privacy-preserving training by rewinding and freezing the privacy-vulnerable and learnability-important weights.
>
> (iii)  Lastly, we would like to underscore that \[1\] is not validated against modern attacks such as, LiRA and RMIA, which are widely recognized as the most powerful empirical proof of membership privacy approaches.  Besides, the most commonly convincing metrics of MIAs such as AUC and TPR@(Low FPR) are also not evaluated in \[1\], although our paper includes all of them.
>
> Therefore, these factors make our study much more comprehensive and convincing. We will refer to \[1\] in our paper and make these points clear. Also, we included new results for \[1\] in Tab.3 of the updated manuscript.

---

> ### Author Response · Authors · 2025-11-24
> **Response - Part 3/3**
>
> **Questions:**
>
> ---
>
> \[Q1\]
> For Relaxloss, we tune its hyper-parameter α in the interval \[0.5, 2.5\]. As for RMIA, we follow the RMIA paper’s recommendation \[5\] to set γ to 2\. We will add a description about hyper-parameter searching and the settings of all approaches. Please let us know if you have any further suggestions regarding this matter. Thank you for your comment.
>
> ---
>
> \[Q2\]
> Although we are not sure if we understood your question correctly, with our best understanding, we think you meant the shadow model technique in LiRA \[4\] and RMIA \[5\] (in the algorithm description of their papers, they call the shadow model using the term “reference model”), which we basically employed. To train target models, we randomly sample data points from the original train set (described in Sec.B of the appendix). To train shadow models, we randomly sample data points from the original train set for the same count as the count of training data points for the target model (a.k.a., victim model in some studies). Then, to get the training data for the “IN” and “OUT” models, we add and delete target examples from the sampled datapoints, respectively.  Please correct us if we misunderstood your question. We will follow up.
>
> ---
>
> **Reference**
>
> \[1\] Hu, Qiang, et al. "Defending membership inference attacks via privacy-aware sparsity tuning." ArXiv. 2024\.
> \[2\] Kaya, Yigitcan, et al. "On the effectiveness of regularization against membership inference attacks." ArXiv. 2020\.
> \[3\] Jia, Jinghan, et al. "Model sparsity can simplify machine unlearning." NeurIPS. 2023\.
> \[4\] Carlini, Nicholas, et al. "Membership inference attacks from first principles." IEEE S\&P. 2022\.
> \[5\] Zarifzadeh, Sajjad, et al. "Low-Cost High-Power Membership Inference Attacks." ICML. 2024\.

---

> > ### Comment · Reviewer_wcwv · 2025-11-26
> >
> > Thank you for your reply. While you addressed some points, I still have the following concerns:
> > 1. Neither Tab. 2 nor Fig. 5 simultaneously presents the privacy metrics (e.g., AUC or TPR at low FPR) alongside test accuracy.
> > 2. There is no ablation study analyzing the impact of the hyperparameter $\lambda$.
> > 3. Why were 18000 samples specifically chosen for training on CIFAR-10/100? Could you provide a table detailing the dataset splits and the usage of each partition?
> > 4. Given that CIFAR-10 and CIFAR-100 share the same total dataset size, what is the justification for using 4000 reference samples for CIFAR-100 but only 2000 for CIFAR-10?
> > 5. What is the specific function of the reference data in Tab. 4, given that LiRA does not traditionally rely on it?

---

> > > ### Author Response · Authors · 2025-12-02
> > >
> > > **\[Q1\]** To present privacy metrics alongside test accuracy, we change the rewinding rates and show the results below:
> > >
> > > | Rewinding Rate | 3% | 5% | 10% |
> > > | :---- | :---- | :---- | :---- |
> > > | Test Accuracy (%) | 77.90 | 76.65 | 76.33 |
> > > | Train Loss | 0.4955 | 0.9096 | 1.6138 |
> > > | Test Loss | 1.0436 | 1.4178 | 1.8352 |
> > > | LiRA TPR(%)@0.1%FPR | 0.55 | 0.42 | 0.18 |
> > >
> > > Good privacy-utility trade-offs can be seen alongside the rewinding rate.
> > >
> > > ---
> > >
> > > **\[Q2\]** λ is to balance the learning and unlearning objective functions. Since training difficulties of various datasets vary a lot, λ needs to be set in different values to avoid fluctuations of the unlearning process that is described in Sec.4.1. The value range is \[0.5, 1.0).
> > >
> > > ---
> > >
> > > **\[Q3\]** The results are essentially based on the study \[2\], to execute shadow model based attacks, sample data points need to be sampled from the original train set for training target models and shadow models, and the data points for shadow models should not heavily overlap.
> > >
> > > ---
> > >
> > > **\[Q4\]** As for more reference data in CIFAR-100 than that in CIFAR-10, it is because the number of samples for each class in CIFAR-100 is too fewer than that in CIFAR-10 when they sample the same quantity, which could affect the estimation process.
> > >
> > > ---
> > >
> > > **\[Q5\]** The reference data does not denote the data for shadow model, which is called reference model in paper \[1\]. It is for the estimation process. Please kindly refer to Sec.4.1 and Fig.2 in the manuscript.
> > >
> > > ---
> > >
> > > **Reference**
> > > \[1\] Carlini, Nicholas, et al. "Membership inference attacks from first principles." IEEE S\&P. 2022\.
> > > \[2\] Chen, Dingfan, et al. "RelaxLoss: Defending Membership Inference Attacks without Losing Utility." ICLR, 2022\.

---

### Author Response · Authors · 2025-12-02
**General Response for Area Chair**

Dear Area Chair:

We would like to provide a summary of our submission and rebuttal. We hope it can help you learn about our paper and contributions quicker and easier.

**Summary of the paper:**
This paper found out that learnability and privacy vulnerability are entangled in a few critical weights, and the learnability of a weight in a neural network is determined by its position rather than its value (magnitude & sign). Based on these insights, we developed a privacy-preservation approach — critical weights rewinding and fine-tuning (CWRF). The effectiveness of our approach is verified by comprehensive experiments.

**Summary of the Rebuttal Discussion:**
We addressed reviewers’ main concerns case by case for experimental evaluation, theoretical analyses, and writing aspects. (i) For experimental evaluations, we addressed the reviewers’ – wcwv, fnse, wW66, and Jji4 –  concerns regarding the strength of MIAs and more evaluation results, including privacy-utility curves, more shadow models, and new NLP domain dataset. (ii) For theoretical analyses, we discussed our study and \[1\] with the reviewers wcwv and Jji4 from multiple angles. Besides, we also addressed other theoretical concerns from the reviews’ – fnse, wW66, and Jji4 – regarding privacy onion effect \[2\] and weight-level Leave-One-Out (LOO) approximation \[3\]. (iii) For writing aspects, we addressed the reviewers’ — fnse, XTZA, and wW66 – suggestions to enhance our paper’s readability. After meaningful and profound discussions, the reviewers fnse, wW66, and Jji4, decided to raise their scores based on our responses.

We will be happy to answer any further questions you may have. Thank you for your time and effort in reviewing our manuscript.

**Reference**
\[1\] Hu, Qiang, et al. "Defending membership inference attacks via privacy-aware sparsity tuning." ArXiv. 2024\.
\[2\] Carlini, N. et al. The Privacy Onion Effect: Memorization is Relative. NeurIPS 2022\.
\[3\] Molchanov, Pavlo, et al. "Importance estimation for neural network pruning." CVPR. 2019\.

---

### Meta-Review · Area_Chair_Kd4B · 2026-01-07

**Summary:**

This paper introduces a new defence method, critical weights rewinding and fine-tuning (CWRF), that curates only privacy-vulnerable weights to defend against membership inference attacks. Reviewers raised concerns about the limited empirical validation on small-scale models and datasets, the lack of hyperparameter sensitivity analysis, weak theoretical motivation, and insufficient experimental evidence regarding vulnerable weights. In the rebuttal, the authors added new experiments, including additional baselines and ablation studies, to strengthen the conclusions presented in the paper. The authors further clarified key insights and motivation behind their method. The reviewers actively participated in the discussion, and three reviewers agreed to increase the scores. By checking the authors’ responses and the reviewers’ feedback, I believe that most concerns have been addressed.

**Reviewer Concerns:**

1. Reviewer wcwv raised major concerns about the lack of sensitivity analysis for the hyperparameter and limited empirical validation on small-scale models and datasets. The authors clarified the ablation experiment of hyperparameters described in the paper. Regarding LLM tasks, the authors do not provide LLM experimental results, noting that this requires further investigation. The reviewer acknowledged that the authors had addressed some of the concerns.

2. Reviewer fnse argued that the paper lacks theoretical motivation and shows insufficient experimental evidence regarding vulnerable weights. The authors acknowledged that the paper is primarily empirical, yet still valuable. The authors added experiments to show the effectiveness of the method when applied to other defence methods. The reviewer noted that the authors addressed key weaknesses and raised the review score.

3. Reviewer XTZA had already recommended acceptance prior to the rebuttal and had only minor suggestions about the manuscript. The authors revised the manuscript according to reviewers' suggestions.

4. Reviewer wW66 raised concerns regarding the experimental details and the justification for the experiments. The authors further clarified the experimental setup described in the manuscript and included experiments to demonstrate the effectiveness of the method.

Reviewer Jji4 initially has a negative attitude towards this paper. The reviewer raised major concerns regarding the soundness of the method and the lack of sufficient experimental validation. The authors clarified the key insights and motivation behind their method and added new experiments to strengthen the conclusions. The reviewer noted that the authors addressed some concerns and increased the score.

**Reviewer Scores:**

Reviewer wcwv may maintain the initial score (Rating: 6).
Reviewer fnse had agreed to raise the score and may increase it from 4 to 6.
Reviewer XTZA noted that the initial score already reflects support for the paper's acceptance (Rating: 8).
Reviewer wW66 had agreed to raise the score and may increase it from 6 to 8.
Reviewer Jji4 also agreed to raise the score and may increase it from 2 to 4.

---

### Decision · Program_Chairs · 2026-01-26

Accept (Poster)